# DBES: A Systematic Benchmark and Metric Suite for Evaluating Expert Specialization in Large-Scale MoEs

## Abstract

Mixture-of-Experts (MoE) architectures scale LLM capacity efficiently, however, a fundamental disconnect remains between their architectural intent and the empirical verification of expert specialization. Conventional evaluations primarily utilize aggregate routing statistics, which often obfuscate whether experts develop genuine, context-aware proficiency or merely act as redundant computational units. To bridge this gap, we introduce a rigorous, multi-faceted framework for assessing MoE specialty. We first construct Domain Bench for Experts Specialty(DBES), a multi-domain benchmark derived from open-source and real-world application data. We then propose a suite of novel metricsincluding Routing Specialization, Normalized Effective Rank, Domain Isolation, Rademacher Complexity, and N-gram Routing Ratio and Group N-gram Ratioto analyze expert behavior at both token and sequence levels. These metrics move beyond load distribution to quantify the structural and functional specialization of experts. We apply this framework to conduct a comprehensive analysis of prominent MoE models, including Qwen3-MoE, DeepSeek-R1, and GLM-4.6. Our experiments reveal distinct specialization strategies: Qwen-series models exhibit high specialization and domain isolation, while DeepSeek and GLM employ more generalized, collaborative expert usage. However, all models show limited n-gram expertise($n = 10$), falling between $0.12\%$ and $1.67\%$. Furthermore, we establish a positive correlation between architectural specialization and performance on complex, domain-specific downstream tasks. This work provides the first systematic methodology for evaluating and interpreting expert specialty, offering crucial insights

for the design, model selection, and efficient post-training of next-generation MoE systems.

## 1. Introduction

The necessity for expert specialization arises from the "Single Token Fallacy," where models fail to disambiguate contexts like "version 9.11" (a software release string) versus "9.11" (a numerical decimal). This homogenization is fundamentally a failure of the gating network to minimize the systems energy across distinct semantic manifolds, causing experts to converge toward generic, redundant solutions rather than specialized proficiency. From an Energy-Based perspective, if the router cannot establish deep energy minima for individualized sub-tasks, the MoE structure effectively collapses into a dense model with high parameter redundancy, nullifying the architectural intent of conditional computation.This structural failure is directly measurable via Rademacher Complexity ($\widehat{\mathcal{R}}_m$). A specialized router demonstrates lower complexity because its gating policy is "stiff"it resists aligning with random noise in favor of "locking" onto the distinct feature manifolds of specific domains. For the "9.11" problem, expertise manifests as a narrowing of the active hypothesis space, where the router minimizes $\widehat{\mathcal{R}}_m$ by establishing stable, low-entropy paths for structured patterns. By linking this complexity to our sequence-level N-gram Expertise (NGR), we move beyond aggregate statistics to a theoretically grounded quantification of how experts transition from generalist units to precision-targeted specialists.

## 2. Related Work

**Token-Level Heuristics and the "Single Token Fallacy".** Early MoE interpretability relied on routing frequency and load distribution (Jiang et al., 2024). LLaMA-MoE utilized L2 distances of routing distributions to quantify domain similarity, while later studies correlated activation with Part-Of-Speech tags (Antoine et al., 2025). However, these metrics often conflate semantic expertise with syntactic regularity. Recent evidence suggests that routing counts alone are insufficient; norm-aware routing analyses (Lo et al., 2025) reveal that output-norm and routing score

---

[1]Anonymous Institution, Anonymous City, Anonymous Region, Anonymous Country. Correspondence to: Anonymous Author <anon.email@domain.com>.

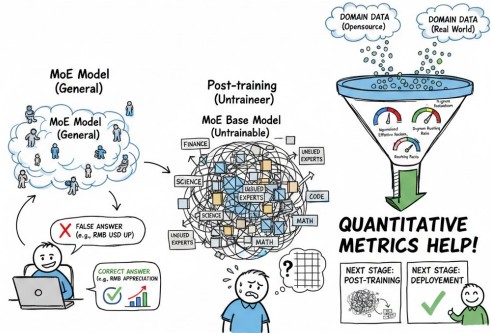

*Figure 1.* **Architectural challenges in MoE and our proposed intervention.** Conventional routing suffers from homogenization due to the "Single Token Fallacy," where experts fail to disambiguate contextual nuances. Our framework fosters specialized expert manifolds and utilizes the Domain Bench for Expert Specialty (DBES) to quantitatively assess functional proficiency and structural isolation.

correlations provide a more accurate proxy for an experts actual contribution to the final residual stream than mere frequency.

**Structural Specialization and Super Experts.** To move beyond purely learned (and often unstable) routing, recent architectures enforce specialization structurally. DeepSeek-MoE (Dai et al., 2024) utilizes shared experts to isolate common knowledge. A pivotal discovery in this domain is the identification of Super Experts (SEs) (Yang et al., 2025; Su et al., 2025), where global-batch load balancing reveals a power-law distribution of expertise. Ablation studies show that removing a few SEs triggers catastrophic model collapse, though such "mask-and-test" methods remain black-box evaluations that fail to explain the semantic nature of the expertise.

**Sequence Dynamics and Stability.** Human language is inherently contextual, rendering single-token analysis (the "Single Token Fallacy") prone to polysemy errors. To address this, "Engram" and MoLE (Jie et al., 2025) architectures utilize $n$-gram indices to ensure experts process complete concepts. SeqTopK (Wen et al., 2025) further stabilizes routing by optimizing cumulative scores over sequences. For consistent evaluation, the SAFEx framework (Lai et al., 2025) introduces stability-based expert analysis, providing a benchmark for how reliably experts respond to long-range temporal dependencies rather than immediate lexical triggers.

**Latent Geometry and Interpretability.** Modern quantification has shifted toward continuous geometric analysis. Liu et al. (2025) and Ying et al. (2025) research into latent spaces to monitor "Rank Collapse" (Dong et al., 2021). Tools like Router Lens (Bai et al., 2025) and Expert Tran-

sition Graphs (CoE) (Wang et al., 2025) allow for the visualization of distinct clusters in the latent manifold. Our work unifies these perspectives by integrating $n$-gram expertise with latent-rank metrics, establishing a theoretically grounded framework for verifying domain-specific functional isolation.

## 3. Analyze methodology

To quantitatively dissect the expert routing behaviors across different domains, we introduce a set of metrics designed to evaluate MoE models. For a given model and a specific domain dataset $D$, we record the routing decisions for every token generated during inference. Let $L$ be the number of layers and $E$ be the number of experts in each MoE layer. We aggregate these decisions to form an Expert Activation Matrix $A \in \mathbb{R}^{L \times E}$, where each entry $A_{ij}$ represents the total count of tokens assigned to the $j$-th expert in the $i$-th layer. To eliminate Expert size $T$ and Top-$k$ influence, we normalized the matrix $M$

$$Mij = \frac{\frac{A_{ij}}{\sum_{m=1}^{T} A_{im}}}{k/T} = \frac{T \cdot A_{ij}}{k \cdot \sum_{m=1}^{T} A_{im}} \qquad (1)$$

### 3.1. Routing Specialization $S_{spec}$

Routing Specialization quantifies the intensity of preference for specific experts within a single domain, calibrated against the model's capacity. To compute this, we first normalize the expert activation matrix $M$ to obtain probability distribution of expert activation, $P_{row}^{(i)}$:

$$P_{\text{row}}^{(i)} = \frac{M_i}{\sum_{j=1}^{E} M_{ij}} \qquad (2)$$

where $M_i$ is the $i$-th row vector of $M$. $S_{spec}$ is then defined as the mean Kullback-Leibler (KL) divergence between this empirical distribution $P_{row}^{(i)}$ and a uniform distribution $Q_{uniform}$ (where $Q_j = \frac{1}{E}$ for all $j$):

$$S_{spec} = \frac{1}{L} \sum_{i=1}^{L} D_{KL} \left( P_{\text{row}}^{(i)} || Q_{\text{uniform}} \right) \qquad (3)$$

This metric quantifies the intensity of a model's preference for specific experts within a single domain. $S_{spec}$ measures the information gain of the learned routing policy leveraging Kullback-Leibler (KL) divergence(Cover & Thomas, 2006). In the MoE literature, this is often discussed inversely in the context of load balancing (Fedus et al., 2022; Lewis et al., 2021), where high entropy (low specialization) is typically enforced to maximize hardware utilization. Here, we use $S_{spec}$ to positively identify the emergence of domain-specific expertise.

## 3.2. Normalized Effective Rank $R_{eff}$

Normalized Effective Rank measures the complexity and variability of expert combination patterns required by a domain. It is derived from the singular values($\sigma_k$) of the probability-normalized matrix $P_{row}$, scaled by the theoretical maximum rank to allow for cross-model comparison:

$$R_{eff} = \frac{1}{min(L, E)} \frac{(\sum \sigma_k)^2}{\sum \sigma_k^2} \tag{4}$$

This is derived from the singular value spectrum of the routing matrix. Originally defined by Roy & Vetterli (2007) as a measure of effective dimensionality, this metric serves as a proxy for the diversity of the routing subspace. A high $R_{eff}$ indicates diverse, non-redundant expert combinations, whereas a low value signals "Rank Collapse"(Dong et al., 2021), where experts exhibit highly correlated activation patterns.

## 3.3. Domain Isolation $S_{iso}$

Domain Isolation evaluates the orthogonality of expert usage between different domains. Let $\mathbf{P}^{(A)}$ denote the matrix of normalized routing probabilities for domain $A$, constructed by stacking the row vectors $P_{row}^{(i)}$ (from Eq 2) for all layers $i = 1 \ldots L$. The domain isolation score is calculated as:

$$S_{iso}(A) = 1 - \frac{1}{N-1} \sum_{B \neq A} \text{Sim}(\mathbf{P}^{(A)}, \mathbf{P}^{(B)}) \tag{5}$$

where $N$ is the number of domains and $Sim(A, B)$ denotes the cosine similarity between the routing probabilities matrix of domains $A$ and $B$.

This metric measures the orthogonality of expert usage across domains. High isolation indicates "Modular" behavior (Pfeiffer et al., 2023), where distinct tasks utilize disjoint expert sets. This decoupling is critical for mitigating negative transfer in multi-task learning(Ma et al., 2018), distinguishing specialized function from shared usage.

## 3.4. Empirical Rademacher-based Routing Complexity $\widehat{\mathcal{R}}_m(\mathbb{G}_{\theta^*})$

Standard Rademacher Complexity typically measures the capacity of a function class to fit arbitrary noise. However, to evaluate the specialization of a converged MoE model, we introduce the **Empirical Rademacher-based Routing Complexity** as a diagnostic metric. Instead of assessing the potential capacity of the entire hypothesis space $\mathcal{G}$, we quantify the "stiffness" of the learned gating manifold $\mathbb{G}_{\theta^*}$. A highly specialized model should be strongly anchored to semantic features and exhibit minimal alignment with stochastic perturbations.

**Definition 3.1** (Empirical Rademacher-based Routing Complexity). Let $\mathbb{G}_{\theta^*} : \mathbb{R}^d \rightarrow \mathbb{R}^E$ be a pre-trained gating policy with fixed parameters $\theta^*$. Given a sample $\mathcal{S} = \{x_1, \ldots, x_m\}$, the empirical routing complexity is defined as the expected alignment between the static routing distribution and a random Rademacher vector $\boldsymbol{\sigma}$:

$$\widehat{\mathcal{R}}_m(\mathbb{G}_{\theta^*}) = \mathbb{E}_{\boldsymbol{\sigma}}\left[\frac{1}{m \cdot E} \sum_{i=1}^{m} \sum_{e=1}^{E} \sigma_{i,e} \cdot \text{softmax}(\mathbb{G}_{\theta^*}(x_i))_e\right] \tag{6}$$

where $\sigma_{i,e}$ are independent Rademacher variables taking values in $\{+1, -1\}$. In practice, we estimate $\widehat{\mathcal{R}}_m$ using $N = 1000$ Monte Carlo iterations.

We distinguish this metric from optimization-based complexity measures. For a post-training appraisal, it serves as a robust proxy for **Routing Determinism**. In an unspecialized or randomly initialized model, the typically $\widehat{\mathcal{R}}_m \approx 10^{-2}$. Conversely, a specialized expert manifold is strictly constrained by learned semantic priors, yielding converged $\approx 10^{-5}$ observed in our experiments. This contrast validates that well-trained MoE model converges to a specialized state.

## 3.5. N-gram Expertise $\mathbb{E}_{\text{NGR}}^{(n)}$

In models where $k > 1$, the routing decision at time $t$ is a set of expert indices $\mathcal{K}_t = \{e_{t,1}, \ldots, e_{t,k}\}$. To maintain mathematical rigor, we clarify the definition of consistency across a trajectory.

**Definition 3.2** (Observed $n$-gram Path Space). Let $\mathcal{X}$ be the set of tokens. For a routing policy $\mathbb{G}$, let $\Omega^{(n)}$ denote the multiset of all $n$-length sliding windows of routing sets observed during inference:

$$\Omega^{(n)} = \{\omega_t \mid \omega_t = (\mathcal{K}_t, \mathcal{K}_{t+1}, \ldots, \mathcal{K}_{t+n-1})\}_{t=1}^{T-n+1} \tag{7}$$

The use of a sliding window ensures that the metric captures the maximum statistical density of the routing manifold, accounting for all local transitions rather than disjoint segments.

**Definition 3.3** (N-gram Expertise for $k \geq 1$). The N-gram Expertise $\mathbb{E}_{\text{NGR}}^{(n)}$ is defined as the empirical probability that at least one expert remains consistently active throughout the $n$-gram window:

$$\mathbb{E}_{\text{NGR}}^{(n)} = \frac{1}{|\Omega^{(n)}|} \sum_{\omega \in \Omega^{(n)}} \mathbb{I}\left(\bigcap_{j=1}^{n} \mathcal{K}_j \neq \emptyset \mid \omega\right) \tag{8}$$

where $\bigcap_{j=1}^{n} \mathcal{K}_j$ is the intersection of the expert sets within the window $\omega = (\mathcal{K}_1, \ldots, \mathcal{K}_n)$.

To validate the definition above, we need to use multiple views to consider a MoE system.

- MoE Expert Choice as Bellman Optimization

  **Lemma 3.4** (Spectral Bound of Bellman Rank in

MoE). *Let* $\mathbf{G} : \mathcal{X} \to \mathbb{R}^E$ *be a C-Lipschitz gating mechanism mapping an input manifold $\mathcal{M}_D$ of intrinsic dimension d to an expert selection space. If the routing process is treated as a sequential decision task, the MoE architecture constitutes a* **Bellman Optimization System** *with a Bellman Rank $\kappa$ bounded by the covering number of the feature manifold, such that* $\kappa \leq \mathcal{O}(d \log C)$.

A formal proof is provided in Appendix C.The feasibility of MoE relies on the existence of a low-rank path, but its measurability is found in the 'n-gram single path' of the selection distribution. In the MoE architecture, the transition to $k$ experts represents a deliberate narrowing of the hypothesis space.

- MoE Expert Choice as Maximum Likelihood Estimation

**Theorem 3.5** (NGR-Mutual Information Relation). *Let $\{p_t\}_{t=1}^T$ denote the sequence of expert indices and $\{x_t\}_{t=1}^T$ the input tokens. Define the n-gram mutual information as $I_n(x; p) = I(x_{t:t+n-1}; p_{t:t+n-1})$.*

*Let $H_{\max} = \log E^n$ be the maximum entropy of the routing path. There exists a constant $C_n$ such that:*

$$\mathbb{E}_{\mathrm{NGR}}^{(n)} \geq \frac{1}{E^{n-1}} \exp\left(\frac{I_n(x; p) - H\max}{n}\right) \quad (9)$$

Proof is also provided in Appendix C. During training, viewed as maximum likelihood estimation, this procedure will force $H(p|x) \to 0$, meaning the router has "learned" its specialty. If the router is random, NGR is low and MI is low.

## 3.6. Group N-gram Expertise $\mathbb{G}_{\mathrm{G\text{-}NGR}}^{(n)}$

Let's define a group expertise since we want to know whether the pack can be speciliazed. Denote $\Gamma^{(n)}$ is the multiset of all $n$-length group-routing trajectories observed in the hierarchical routing manifold.

**Definition 3.6** (Group N-gram Expertise for $k \geq 1$). Let $\mathcal{G}_j = \{\mathrm{Group}(e) : e \in \mathcal{K}_j\}$ be the set of functional group indices activated at position $j$. The Group N-gram Expertise $\mathbb{G}_{\mathrm{G\text{-}NGR}}^{(n)}$ measures the functional stability within the group hierarchy across an $n$-length sliding window:

$$\mathbb{G}_{\mathrm{G\text{-}NGR}}^{(n)} = \frac{1}{|\Gamma^{(n)}|} \sum_{\gamma \in \Gamma^{(n)}} \mathbb{I}\left(\bigcap_{j=1}^n \mathcal{G}_j \neq \emptyset \mid \gamma\right) \quad (10)$$

where $\Gamma^{(n)}$ is the multiset of all $n$-length group-routing trajectories $\gamma = (\mathcal{G}_1, \ldots, \mathcal{G}_n)$ observed in the corpus, and $\bigcap_{j=1}^n \mathcal{G}_j$ denotes the intersection of group sets within the window.

This metric is calculated on multi-experts for the specialized domain application, theoretical validation can refer to Appendix C.

## 4. Experiments

### 4.1. Setup

We conduct a comprehensive evaluation of expert routing behaviors across several mainstream MoE models, including Qwen3-30B (Instruct & Thinking), Qwen3-235B-Thinking, GLM-4.6, and DeepSeek-R1. To quantify the domain-specific expertise of these models, we focus on analyzing the suite of metrics in Section 3.

**Dataset**. to validate the expertise in different domain, we establish a database from open-source dataset of seven different domain with 9 partitions from different source. Table8 shows statistics of Domain Bench for Expert Specialty(**DBES**).This benchmark aggregates diverse cognitive tasks to rigorously assess expert specialization. It spans logical reasoning (AIME 2025, Yale-FinanceMath), professional knowledge (BigBio MedQA, Nguha LegalBench), and scientific literacy (AllenAI SciQ), while also distinguishing between standard coding tasks (LiveCodeBench) and complex software engineering (Princeton SWE-bench). Details of the Data Bench is in Appendix E.

**Experiment Setting**. On this database, experiments are conducted in an H20 environment, with inference implemented via SGLang integrated with our custom evaluation metrics. To ensure full reproducibility of our work, we release open-source datasets on HuggingFace(Moe-lab/DBES), code repositories on GitHub (*MoE-Evaluator/Specialization_Metrics_of_MoE*) and additional implementations in the supplementary materials, with the generation hyperparameters set as follows: $top - k = 1$, $temperature = 0$, and $max - tokens = 1024$.

We evaluate our framework across the full spectrum of the DBES benchmark. To maintain a focused discussion on the macro-patterns of expert specialization, we present a synthesized analysis of our core metrics here. However, to support deep interpretability and reproducibility, we provide exhaustive layer-wise activation heatmaps for all architectures (94 layers for Qwen-235B) and detailed per-partition data statistics in Appendix. This supplemental repository offers a granular diagnostic of the routing manifolds and confirms the stability of the expertise phenomena across diverse data distributions.

### 4.2. Results and Metrics Analysis

#### 4.2.1. ROUTING SPECIALIZATION

Routing Specialization($S_{spec}$)measures the deviation of the expert activation distribution from a uniform distribu-

*Table 1.* Comprehensive analysis of Domain Isolation ($S_{iso}$), Routing Specialization ($S_{spec}$), and Normalized Effective Rank ($R_{eff}$) across different domains.

| MODEL | METRIC | MATH | SCIENCE | MEDICAL | MEDICAL2 | KNOWLEDGE | CODE | LEGAL | CODE2 | FINANCE |
|---|---|---|---|---|---|---|---|---|---|---|
| | $S_{spec}$ | 0.63 | **0.80** | 0.90 | 0.91 | 0.43 | 0.44 | 0.95 | 0.63 | 0.57 |
| QWEN3-30B-INSTRUCT | $R_{eff}$ | 0.44 | 0.45 | 0.50 | 0.50 | 0.37 | 0.39 | 0.47 | 0.44 | 0.41 |
| | $S_{iso}$ | **0.56** | **0.54** | 0.57 | 0.57 | **0.44** | **0.54** | **0.64** | **0.62** | **0.58** |
| | $S_{spec}$ | 0.67 | 0.79 | **1.09** | **1.09** | 0.47 | **0.64** | **0.98** | **0.76** | **0.62** |
| QWEN3-30B-THINKING | $R_{eff}$ | 0.44 | 0.45 | 0.54 | 0.54 | **0.39** | 0.46 | 0.48 | 0.46 | 0.42 |
| | $S_{iso}$ | 0.55 | 0.43 | **0.58** | 0.58 | 0.38 | 0.41 | 0.53 | 0.55 | 0.52 |
| | $S_{spec}$ | **0.74** | 0.70 | 0.99 | 0.99 | **0.49** | **0.64** | 0.94 | **0.76** | 0.57 |
| QWEN3-235B-THINKING | $R_{eff}$ | 0.28 | 0.26 | 0.31 | 0.31 | 0.24 | 0.28 | 0.27 | 0.27 | 0.26 |
| | $S_{iso}$ | 0.51 | 0.45 | **0.58** | 0.58 | 0.37 | 0.44 | 0.56 | 0.57 | 0.51 |
| | $S_{spec}$ | 0.24 | 0.24 | 0.31 | 0.31 | 0.10 | 0.21 | 0.31 | 0.23 | 0.22 |
| DEEPSEEK-R1-0528 | $R_{eff}$ | **0.49** | **0.47** | 0.54 | 0.54 | 0.28 | **0.46** | **0.56** | **0.49** | 0.47 |
| | $S_{iso}$ | 0.36 | 0.31 | 0.31 | 0.31 | 0.22 | 0.36 | 0.39 | 0.40 | 0.33 |
| | $S_{spec}$ | 0.35 | 0.33 | **0.47** | 0.47 | 0.15 | 0.30 | 0.41 | 0.23 | 0.33 |
| GLM-4.6 | $R_{eff}$ | **0.49** | **0.47** | **0.59** | **0.59** | 0.30 | **0.46** | 0.54 | 0.42 | **0.51** |
| | $S_{iso}$ | 0.42 | 0.31 | 0.39 | 0.39 | 0.26 | 0.34 | 0.38 | 0.39 | 0.37 |

tion (based on KL divergence) when processing domain-specific tasks. Experimental results (Table.1) reveal distinct strategic differences among models:

- **Strong Specialization in Qwen Series:** Qwen models demonstrate high routing specialization, peaking in Qwen3-235B-Thinking (Medical:0.99, Legal:0.94). This confirms that scaling enables a highly sparse routing strategy, effectively modularizing knowledge storage.

- **Reinforcement Effect of Thinking Mode:** Comparing Qwen3-30B-Instruct with Qwen3-30B-Thinking, we observe CoT fine-tuning consistently boost $S_{spec}$ (e.g., Medical rising from 0.90 to 1.09). This suggests that complex reasoning tasks demand precise routing to specific expert subsets.

- **Distributed Strategy in DeepSeek/GLM:** In contrast, DeepSeek-R1 and GLM-4.6 exhibit significantly lower specialization. Their distributed representation strategy favors collaborative expert activation, prioritizing robustness over the functional decoupling seen in Qwen.

#### 4.2.2. NORMALIZED EFFECTIVE RANK

Normalized Effective Rank($R_{eff}$) reflects the linear independence and complexity of expert routing patterns. Data analysis, as summarized in Table.1, highlights the trade-off between parameter scale and routing efficiency:

- Qwen3-235B exhibits significantly lower $R_{eff}$ than other models, typically ranging between 0.24 and 0.31. We assume that massive expert redundancy exists within giant MoE models. Although the number of physical experts is vast, they often form fixed

"cliques" to work collaboratively, leading to a collapse in effective routing dimensions. This is a typical characteristic of trading "parameter redundancy" for "memory capacity."

- Conversely, Qwen3-30B, DeepSeek, and GLM maintain higher $R_{eff}$ levels (0.45 ~ 0.55). This suggests that the routing mechanisms in these models are more efficient, where expert transformations at each layer provide substantial information gain, demonstrating stronger independence among experts.

- High-Dimensional Utilization in DeepSeek: Despite DeepSeek's low specialization, its effective rank remains high ( 0.49). This corroborates its "distributed" nature: while it activates many experts per step, the combinition of these experts are diverse rather than mechanically repetitive.

#### 4.2.3. DOMAIN ISOLATION

Domain Isolation($S_{iso}$) measures the orthogonality of expert subsets activated by different domain tasks. Our analysis, as summarized in Table.1, indicates:

- Positive Correlation between Domain Decoupling and Expert Specialization: $S_{iso}$ aligns closely with $S_{spec}$. Qwen3-235B and Qwen3-30B achieve high isolation in specialized domains like Medical and Legal, effectively forming distinct expert divisions.

- Architectural Preference for Isolation: DeepSeek and GLM show lower overall isolation (generally < 0.4), indicating a preference for knowledge sharing. While potentially beneficial for cross-disciplinary tasks, this increases interference risks during domain-specific fine-tuning compared to Qwen.

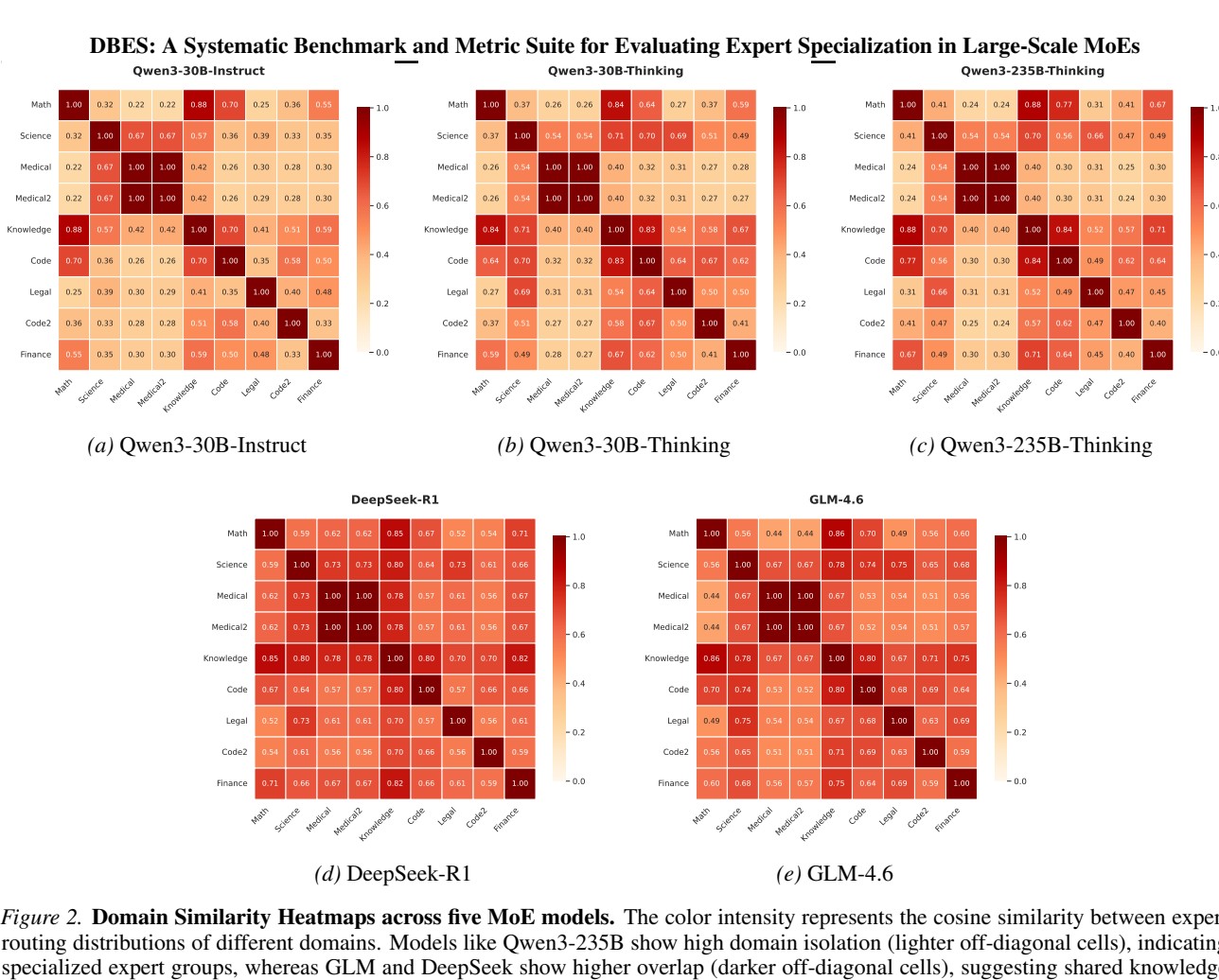

*Figure 2.* **Domain Similarity Heatmaps across five MoE models.** The color intensity represents the cosine similarity between expert routing distributions of different domains. Models like Qwen3-235B show high domain isolation (lighter off-diagonal cells), indicating specialized expert groups, whereas GLM and DeepSeek show higher overlap (darker off-diagonal cells), suggesting shared knowledge representation.

- All models exhibit the lowest isolation in the "Knowledge" domain (Qwen:$0.37 \sim 0.44$, DeepSeek:$0.22$). Given the dataset's multi-disciplinary nature (Math, CS, etc.), this confirms the rationality of our metric.

- Figure 2 confirms these patterns. Qwen shows a sparse structure (distinct experts), whereas DeepSeek and GLM display dense, high-similarity heatmaps across domains.

#### 4.2.4. RADEMACHER COMPLEXITY

- Complexity of Qwen3-30B-Instruct is higher than Qwen3-30B-Thinking means a higher complexity in choosing experts, indicating Qwen3-30B specialty is strengthed by thinking CoT training.

- DeepSeek-R1 shows the largest Rademacher complexity which is comparable to its strategy of selecting 8 out of 256 experts, but still with a far lower complexity compared with other model choice.

- GLM shows the largest Rademacher complexity although it has 160 experts per layer, between Qwen3-moe and DeepSeek; GLM shows limited consistency in achieving expertise from the view of expert Rademacher complexity.

$\widehat{\mathcal{R}}_m(\mathbb{G}_{\theta*})$ values represent the normalized correlation between random noise ($\sigma$) and the model's routing preferences. A value of $10^{-5}$ indicates that the routing manifold is highly specialized and "stiff," meaning it does not easily align with random noisea hallmark of high $n$-gram expertise. Above all, the Rademacher complexity is a weak indication of expertise in MoE models, but it shows that in MoE LLM models, expert choice is far simple than the choice of best sequence, which we will examine latter.

#### 4.2.5. N-GRAM EXPERTISE RATIO $\mathbb{E}_{\text{NGR}}^{(n)}, \mathbb{G}_{\text{G-NGR}}^{(n)}$

To interpret the reported $\mathbb{E}_{\text{NGR}}^{(n)}$ values, we establish a marginal-matched i.i.d. baseline $\mathbb{B}_{\text{NGR}}^{(n)} = \sum_{e=1}^{T}(p_e)^n$, where $p_e$ is the observed empirical activation probability of expert $e$. For $n = 10$, while our recorded expertise ratios fall within $[0.1\%, 1.7\%]$, they are approximately $10^7$ **to** $10^{10}$ **times higher** than the theoretical baseline ($\mathbb{B} \approx 10^{-12}$

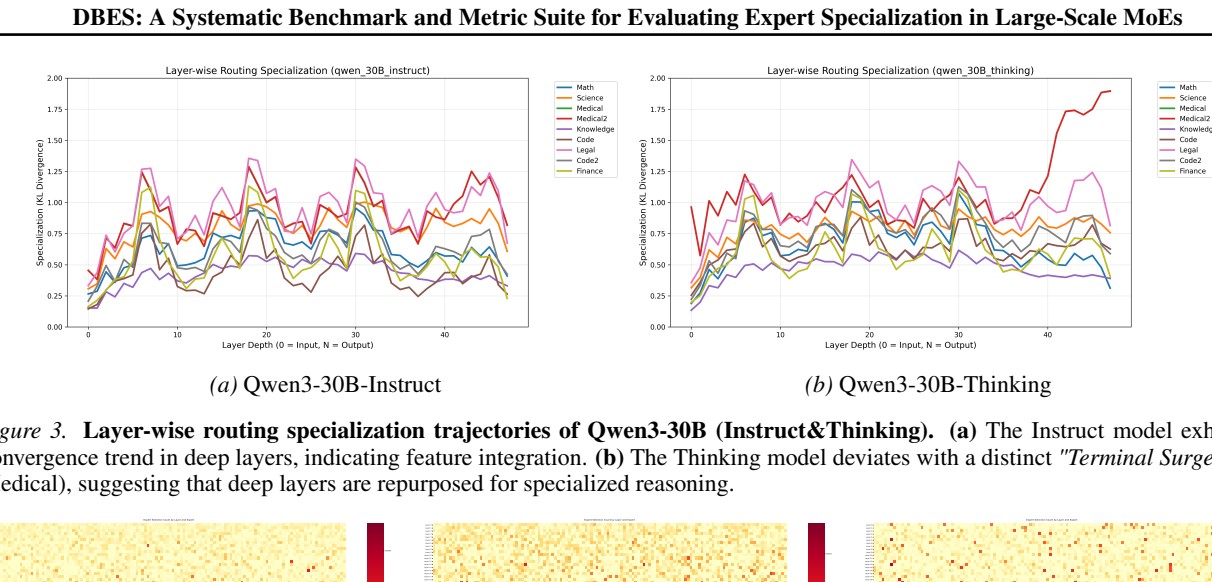

*(a)* Qwen3-30B-Instruct          *(b)* Qwen3-30B-Thinking

*Figure 3.* **Layer-wise routing specialization trajectories of Qwen3-30B (Instruct&Thinking).** **(a)** The Instruct model exhibits a convergence trend in deep layers, indicating feature integration. **(b)** The Thinking model deviates with a distinct *"Terminal Surge"* (e.g., Medical), suggesting that deep layers are repurposed for specialized reasoning.

*(a)* Qwen235B-code heatmap      *(b)* Qwen235B-science heatmap      *(c)* Qwen235B-medical heatmap

*Figure 4.* **Expert activation heatmaps for Qwen3-235B across diverse domains.** The $y$-axis represents the MoE layer index, and the $x$-axis represents the expert index. (a) Coding exhibits sparse, highly specialized activation; (b) Science shows a more uniform distribution; (c) Medical represents a transitional state with localized expertise clusters.

*Table 2.* Rademacher Complexity, N-gram and Group N-gram ratio for different models.

| MODEL | EXPERTS | $\widehat{\mathcal{R}}_m(\mathbb{G}_{\theta^*})$ | $\mathbb{E}_{\text{NGR}}^{(10)}$ | $\mathbb{G}_{\text{G-NGR}}^{(10)}$ |
|---|---|---|---|---|
| QWEN3-30B-THINKING | 128→8 | 2.0018E-05 | 0.69% | 1.30% |
| QWEN3-30B-INSTRUCT | 128→8 | 1.8778E-05 | 0.46% | 2.18% |
| QWEN3-235B | 128→8 | 1.9506E-05 | 0.60% | 2.23% |
| GLM-4.6 | 160→8 | 2.1643E-05 | 0.46% | 0.71% |
| DEEPSEEK-R1-0528 | 256→8 | 1.7433E-05 | 0.73% | 1.39% |

- GLM model shows the second largest n-gram metric, GLM used a higher dimension for MoE experts, which selects 8 out of 160 experts, which is smaller compared with DeepSeek but bigger than Qwen3-moe, GLM show more contracted expert routing space.

- DeepSeek differs for its hierarchical routing strategy, which result in very small Rademacher Complexity, compared with Qwen3-moe series and GLM 4.6. The same reason results in low n-gram expert ratio compared with other models, from Table.2 we see only 0.73% in average.

### 4.3. Ablation Study

Comparison of single metric between different models show us how the model is scored in some metrics then we show the study of different model performance in some fields.

#### 4.3.1. RADEMACHER COMPLEXITY CONVERGECE

To address the estimator's stability, we evaluate the empirical Rademacher Complexity $\widehat{\mathcal{R}}_m(\mathbb{G}_{\theta^*})$ for the Qwen3-30B gating manifold across varying Monte Carlo sample sizes

for uniform routing $128 \rightarrow 8$ MoE). This massive divergence confirms that the observed sequence-level expert consistency is not a statistical artifact of marginal utilization, but a robust manifestation of learned temporal specialization within the routing manifold.

- Qwen3 series show varied expertise in n-gram(NGR) and Group-n-gram(G-NGR), Qwen3-30B, has more specific n-gram routing which will fall into the same experts, while Qwen3-235B showed higher variance.

$(n \in \{500, 1000, 2000\})$. As shown in Table 3, the complexity scores exhibit a robust convergence toward the $10^{-5}$ magnitude. Notably, the observed $O(1/n)$ decay rate suggests that the routing hypothesis space $\mathcal{G}$ is highly structured and concentrated. This negligible complexity relative to a random uniform policy (where $\widehat{\mathcal{R}}_m \gg 10^{-3}$) confirms that the expert selection process is exceptionally "stiff" and less susceptible to stochastic noise. Such low complexity is a theoretical prerequisite for the high N-gram Expertise (NGR) observed in our empirical tests, reflecting a highly specialized and deterministic expert manifold.

*Table 3.* Qwen3-30B-Thinking $\widehat{\mathcal{R}}_m(\mathbb{G}_{\theta^*})$ with different Samples

| Samples | n=500 | n=1000 | n=2000 |
|---|---|---|---|
| $\widehat{\mathcal{R}}_m(\mathbb{G}_{\theta^*})$ | 3.99E-05 | 2.00E-05 | 1.00E-05 |

### 4.3.2. CORRELATION BETWEEN PERFORMANCE AND SPECIALTY

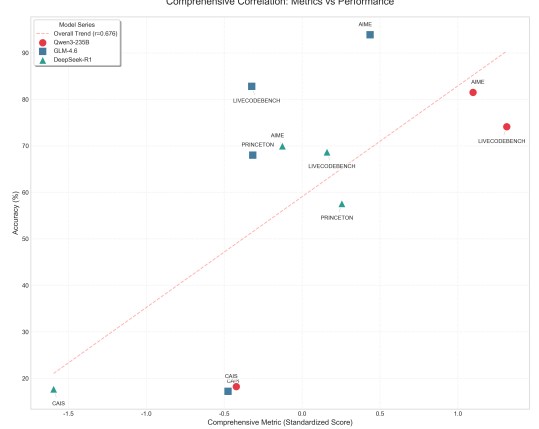

*Figure 5.* **Scaling Specialist Proficiency.** A comprehensive correlation analysis across Qwen3-235B, DeepSeek-R1-0528, and GLM-4.6. The positive linear fit confirms that higher architectural expertise scores are strongly predictive of superior performance in complex reasoning and coding tasks.

To quantify the impact of structural specialization on model utility, we correlate our proposed metrics with objective benchmark performance across the Qwen3, GLM, and DeepSeek architectures.

Figure 5 reveals that as the expertise manifold becomes more isolated and stable, the model's ability to handle specific-domain increases. This positive correlation suggests that the "specialty" of an MoE model serves as a reliable proxy for its generalization ceiling in professional-grade vertical applications. Further analysis is provided in Appendix B.

### 4.3.3. LAYERWISE EXPERTISE

As shown in Figure 3, CoT fine-tuning induces a distinct 'Terminal Surge' in routing specialization. Unlike the Instruct model, which shows feature convergence in deeper layers, the Thinking model exhibits a sharp increase in expert focus for domains like Medical. This indicates that CoT-finetuned models leverage the final layers as a specialized reasoning engine, preventing the collapse of domain features until the very last moment of inference.For a more comprehensive layer-wise analysis covering additional models and domains, please refer to Appendix D.

### 4.3.4. DOMAIN-SPECIFIC EXPERT ALLOCATION

We visualize the expert activation trajectories for Qwen3-235B across representative domains in Figure 4. The heatmaps reveal distinct vertical patterns across layers ($y$-axis) and horizontal concentrations across experts ($x$-axis), reflecting a highly non-uniform distribution. Notably, the model exhibits varying degrees of functional specialization: while the Science domain triggers a relatively balanced activation, the Medical domain shows moderate clustering. In contrast, the Coding domain demonstrates the most aggressive specialization, characterized by sparse, high-intensity activation. Given that Qwen3-MoE omits shared experts, these results confirm that the model has successfully partitioned its capacity into domain-isolated functional units. Comprehensive activation maps for additional domains and architectures are provided in Appendix A.3.

## 5. Conclusion and Future Work

**Conclusion:** We proposed a comprehensive metric suite to evaluate expertise in quantized MoE models. While currently serving as a diagnostic tool, these metrics provide a blueprint for optimizing post-training processes. Our goal is to shift from static evaluation to active optimization, leveraging these indicators to foster deeper functional specialization. **Future work** will focus on:

**Application of Metrics in Post-Training:** We will explore how to effectively utilize these metrics during the post-training phase, particularly in reinforcement learning contexts, to emphasize and enhance the expertise of individual experts, thereby improving performance across various domains.

**Addressing Load Balancing Challenges:** The specialization of experts for specific downstream tasks can lead to load balancing issues. While this specialization is beneficial, it inherently creates challenges in load distribution. To mitigate this, we will investigate additional techniques such as distillation and pruning. Addressing these challenges will be crucial for achieving a stable application of MoE within targeted domains.

## Impact Statement

This paper presents work whose goal is to advance the field of Machine Learning. There are many potential societal consequences of our work, none which we feel must be specifically highlighted here.

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

# A. Experiments Result with Further Analysis

## A.1. Full Table of all metrics

Table 4 shows all metrics of 5 models on 9 dataset (7 domains), with this projected to Domain Expertise, the min/max value is set as bold in Table 5.

*Table 4.* Comparison of Qwen3, DeepSeek-R1-0528, and GLM-4.6 benchmarks.

| Model | Partition | Domain | $S_{spec}$ | $R_{eff}$ | $S_{iso}$ | $\widehat{\mathcal{R}}_m(\mathbb{G}_{\theta*})$-500 | $\widehat{\mathcal{R}}_m(\mathbb{G}_{\theta*})$-1000 | $\widehat{\mathcal{R}}_m(\mathbb{G}_{\theta*})$-2000 | $\mathbb{E}_{NGR}^{(n)}$-2 | $\mathbb{E}_{NGR}^{(n)}$-5 | $\mathbb{E}_{NGR}^{(n)}$-10 | $\mathbb{E}_{NGR}^{(n)}$-20 | $\mathbb{G}_{G\text{-}NGR}^{(n)}$-2 | $\mathbb{G}_{G\text{-}NGR}^{(n)}$-5 | $\mathbb{G}_{G\text{-}NGR}^{(n)}$-10 | $\mathbb{G}_{G\text{-}NGR}^{(n)}$-20 |
|---|---|---|---|---|---|---|---|---|---|---|---|---|---|---|---|---|
| Qwen3-30B-Instruct | aime 2025 | Math | 0.63 | 0.44 | 0.56 | 3.71E-05 | 1.70E-05 | 1.07E-05 | 17.23% | 1.05% | 0.12% | 0.01% | 19.78% | 4.37% | 0.69% | 0.04% |
| | AllenAI SciQ (Val) | Science | 0.80 | 0.45 | 0.54 | 3.76E-05 | 2.15E-05 | 8.83E-06 | 27.47% | 5.70% | 1.46% | 0.24% | 29.00% | 11.01% | 3.78% | 0.81% |
| | BigBio MedQA (Dev) | Medical | 0.90 | 0.50 | 0.57 | 3.74E-05 | 1.97E-05 | 9.63E-06 | 31.98% | 6.94% | 1.67% | 0.23% | 37.76% | 16.87% | 6.39% | 1.52% |
| | BigBio MedQA (Test) | Medical2 | 0.91 | 0.50 | 0.57 | 2.93E-05 | 1.86E-05 | 1.00E-05 | 32.01% | 6.95% | 1.67% | 0.22% | 37.83% | 17.04% | 6.62% | 1.66% |
| | CAIS HLE | Knowledge | 0.43 | 0.37 | 0.44 | 4.77E-05 | 2.38E-05 | 1.16E-05 | 22.01% | 2.57% | 0.57% | 0.17% | 14.40% | 3.59% | 0.80% | 0.12% |
| | LiveCodeBench (Test) | Code | 0.44 | 0.39 | 0.54 | 4.49E-05 | 2.09E-05 | 9.79E-06 | 22.69% | 3.79% | 1.01% | 0.23% | 20.99% | 7.60% | 2.85% | 0.83% |
| | Nguha LegalBench | Legal | 0.95 | 0.47 | 0.64 | 4.14E-05 | 1.85E-05 | 8.75E-06 | 25.69% | 5.31% | 1.50% | 0.31% | 30.67% | 11.92% | 4.09% | 0.86% |
| | Princeton SWE-bench (Test) | Code2 | 0.63 | 0.44 | 0.62 | 3.74E-05 | 2.04E-05 | 1.05E-05 | 22.05% | 2.05% | 0.26% | 0.03% | 19.79% | 5.27% | 1.13% | 0.13% |
| | Yale-FinanceMath (Val) | Finance | 0.57 | 0.41 | 0.58 | 4.60E-05 | 1.98E-05 | 1.05E-05 | 24.89% | 3.40% | 0.60% | 0.08% | 29.10% | 10.46% | 2.98% | 0.53% |
| | Avg. | | 0.70 | 0.44 | 0.56 | 3.99E-05 | 2.00E-05 | 1.00E-05 | 25.11% | 4.20% | 0.98% | 0.17% | 26.59% | 9.79% | 3.26% | 0.72% |
| Qwen3-30B-Thinking | AIME 2025 | Math | 0.67 | 0.44 | 0.55 | 3.82E-05 | 1.92E-05 | 1.04E-05 | 17.18% | 1.31% | 0.15% | 0.01% | 20.02% | 5.14% | 1.00% | 0.09% |
| | AllenAI SciQ (Val) | Science | 0.79 | 0.45 | 0.43 | 4.09E-05 | 1.70E-05 | 9.54E-06 | 21.00% | 2.13% | 0.28% | 0.02% | 19.66% | 4.63% | 0.87% | 0.09% |
| | BigBio MedQA (Dev) | Medical | 1.09 | 0.54 | 0.58 | 3.14E-05 | 1.41E-05 | 9.13E-06 | 27.82% | 4.70% | 0.97% | 0.12% | 37.28% | 15.05% | 5.59% | 1.73% |
| | BigBio MedQA (Test) | Medical2 | 1.09 | 0.54 | 0.58 | 3.15E-05 | 1.76E-05 | 8.80E-06 | 27.84% | 4.72% | 0.99% | 0.13% | 37.23% | 15.07% | 5.65% | 1.76% |
| | CAIS HLE | Knowledge | 0.47 | 0.39 | 0.38 | 4.36E-05 | 2.07E-05 | 1.04E-05 | 20.17% | 2.13% | 0.36% | 0.06% | 13.59% | 2.74% | 0.52% | 0.06% |
| | LiveCodeBench (Test) | Code | 0.64 | 0.46 | 0.41 | 4.80E-05 | 2.06E-05 | 1.17E-05 | 19.13% | 1.99% | 0.34% | 0.04% | 19.77% | 4.92% | 1.16% | 0.18% |
| | Nguha LegalBench | Legal | 0.98 | 0.48 | 0.53 | 3.45E-05 | 1.79E-05 | 7.26E-06 | 22.10% | 2.69% | 0.45% | 0.04% | 24.96% | 7.55% | 1.93% | 0.27% |
| | Princeton SWE-bench (Test) | Code2 | 0.76 | 0.46 | 0.55 | 3.48E-05 | 2.04E-05 | 1.09E-05 | 20.24% | 1.57% | 0.19% | 0.03% | 23.85% | 6.51% | 1.42% | 0.17% |
| | Yale-FinanceMath (Val) | Finance | 0.62 | 0.42 | 0.52 | 4.33E-05 | 2.15E-05 | 9.19E-06 | 22.22% | 2.66% | 0.43% | 0.04% | 24.32% | 6.96% | 1.52% | 0.17% |
| | Avg. | | 0.79 | 0.46 | 0.50 | 3.85E-05 | 1.88E-05 | 9.70E-06 | 21.97% | 2.65% | 0.46% | 0.05% | 24.52% | 7.62% | 2.18% | 0.50% |
| Qwen3-235B-Thinking | AIME 2025 | Math | 0.74 | 0.28 | 0.51 | 3.96E-05 | 1.87E-05 | 9.34E-06 | 19.26% | 1.73% | 0.25% | 0.04% | 21.20% | 5.34% | 1.13% | 0.16% |
| | AllenAI SciQ (Val) | Science | 0.70 | 0.26 | 0.45 | 4.34E-05 | 1.84E-05 | 9.65E-06 | 22.68% | 2.76% | 0.48% | 0.05% | 17.91% | 4.34% | 0.89% | 0.10% |
| | BigBio MedQA (Dev) | Medical | 0.99 | 0.31 | 0.58 | 3.74E-05 | 1.99E-05 | 8.60E-06 | 28.50% | 4.87% | 1.10% | 0.17% | 36.77% | 15.15% | 5.50% | 1.39% |
| | BigBio MedQA (Test) | Medical2 | 0.99 | 0.31 | 0.58 | 3.56E-05 | 1.75E-05 | 8.54E-06 | 28.51% | 4.87% | 1.11% | 0.17% | 36.71% | 15.10% | 5.47% | 1.39% |
| | CAIS HLE | Knowledge | 0.49 | 0.24 | 0.37 | 4.85E-05 | 2.39E-05 | 1.28E-05 | 21.88% | 2.61% | 0.50% | 0.08% | 13.10% | 2.63% | 0.51% | 0.08% |
| | LiveCodeBench (Test) | Code | 0.64 | 0.28 | 0.44 | 3.89E-05 | 1.92E-05 | 9.38E-06 | 21.59% | 2.62% | 0.48% | 0.07% | 21.11% | 5.82% | 1.50% | 0.26% |
| | Nguha LegalBench | Legal | 0.94 | 0.27 | 0.56 | 3.43E-05 | 1.97E-05 | 9.05E-06 | 24.50% | 3.32% | 0.60% | 0.07% | 27.04% | 8.28% | 1.90% | 0.21% |
| | Princeton SWE-bench (Test) | Code2 | 0.76 | 0.27 | 0.57 | 3.89E-05 | 1.90E-05 | 9.62E-06 | 21.41% | 1.71% | 0.19% | 0.02% | 20.57% | 4.92% | 0.86% | 0.08% |
| | Yale-FinanceMath (Val) | Finance | 0.57 | 0.26 | 0.51 | 4.13E-05 | 1.93E-05 | 1.01E-05 | 24.53% | 3.58% | 0.68% | 0.09% | 26.29% | 8.46% | 2.27% | 0.39% |
| | Avg. | | 0.76 | 0.28 | 0.51 | 3.98E-05 | 1.95E-05 | 9.68E-06 | 23.65% | 3.12% | 0.60% | 0.08% | 24.52% | 7.78% | 2.23% | 0.45% |
| DeepSeek-R1-0528 | AIME 2025 | Math | 0.24 | 0.49 | 0.36 | 3.32E-05 | 1.71E-05 | 8.22E-06 | 18.69% | 1.87% | 0.30% | 0.04% | 13.93% | 3.12% | 0.63% | 0.08% |
| | AllenAI SciQ (Val) | Science | 0.24 | 0.47 | 0.31 | 3.45E-05 | 1.75E-05 | 8.54E-06 | 21.23% | 2.88% | 0.64% | 0.09% | 10.67% | 2.60% | 0.66% | 0.11% |
| | BigBio MedQA (Dev) | Medical | 0.31 | 0.54 | 0.31 | 3.42E-05 | 1.68E-05 | 8.34E-06 | 19.43% | 2.45% | 0.48% | 0.06% | 13.08% | 3.18% | 0.78% | 0.11% |
| | BigBio MedQA (Test) | Medical2 | 0.31 | 0.54 | 0.31 | 3.35E-05 | 1.70E-05 | 8.29E-06 | 19.37% | 2.41% | 0.47% | 0.06% | 13.02% | 3.11% | 0.75% | 0.11% |
| | CAIS HLE | Knowledge | 0.10 | 0.28 | 0.22 | 3.70E-05 | 1.92E-05 | 8.96E-06 | 18.60% | 1.95% | 0.37% | 0.06% | 5.87% | 1.00% | 0.21% | 0.03% |
| | LiveCodeBench (Test) | Code | 0.21 | 0.46 | 0.36 | 3.56E-05 | 1.77E-05 | 8.56E-06 | 21.72% | 2.85% | 0.60% | 0.10% | 12.26% | 3.36% | 0.93% | 0.20% |
| | Nguha LegalBench | Legal | 0.31 | 0.56 | 0.39 | 3.40E-05 | 1.76E-05 | 8.21E-06 | 20.51% | 2.36% | 0.40% | 0.04% | 14.24% | 3.73% | 0.83% | 0.10% |
| | Princeton SWE-bench (Test) | Code2 | 0.23 | 0.49 | 0.40 | 3.35E-05 | 1.69E-05 | 8.15E-06 | 22.07% | 2.62% | 0.48% | 0.06% | 14.81% | 3.90% | 0.89% | 0.12% |
| | Yale-FinanceMath (Val) | Finance | 0.22 | 0.47 | 0.33 | 3.49E-05 | 1.71E-05 | 8.32E-06 | 18.82% | 2.00% | 0.39% | 0.07% | 13.28% | 3.21% | 0.75% | 0.13% |
| | Avg. | | 0.24 | 0.48 | 0.33 | 3.45E-05 | 1.74E-05 | 8.40E-06 | 20.05% | 2.38% | 0.46% | 0.06% | 12.35% | 3.02% | 0.71% | 0.11% |
| DeepSeek-R1-0528 | aime 2025 | Math | 0.35 | 0.49 | 0.42 | 3.97E-05 | 2.03E-05 | 1.07E-05 | 19.80% | 1.76% | 0.23% | 0.02% | 17.52% | 4.00% | 0.83% | 0.11% |
| | AllenAI SciQ (Val) | Science | 0.33 | 0.47 | 0.31 | 4.24E-05 | 2.20E-05 | 1.10E-05 | 25.35% | 3.72% | 0.77% | 0.10% | 13.92% | 3.60% | 0.83% | 0.12% |
| | BigBio MedQA (Dev) | Medical | 0.47 | 0.59 | 0.39 | 3.98E-05 | 2.03E-05 | 9.98E-06 | 28.39% | 5.43% | 1.44% | 0.27% | 23.92% | 8.97% | 3.16% | 0.76% |
| | BigBio MedQA (Test) | Medical2 | 0.47 | 0.59 | 0.39 | 4.18E-05 | 2.11E-05 | 1.01E-05 | 28.35% | 5.41% | 1.43% | 0.26% | 23.72% | 8.83% | 3.08% | 0.73% |
| | CAIS HLE | Knowledge | 0.15 | 0.30 | 0.26 | 4.49E-05 | 2.37E-05 | 1.13E-05 | 24.20% | 3.61% | 0.83% | 0.15% | 10.99% | 2.94% | 0.84% | 0.18% |
| | LiveCodeBench (Test) | Code | 0.30 | 0.46 | 0.34 | 4.22E-05 | 2.26E-05 | 1.13E-05 | 22.40% | 2.69% | 0.44% | 0.04% | 12.64% | 2.92% | 0.56% | 0.05% |
| | Nguha LegalBench | Legal | 0.41 | 0.54 | 0.38 | 4.31E-05 | 2.01E-05 | 9.72E-06 | 24.59% | 3.34% | 0.64% | 0.08% | 18.72% | 5.57% | 1.43% | 0.24% |
| | Princeton SWE-bench (Test) | Code2 | 0.23 | 0.42 | 0.39 | 4.50E-05 | 2.25E-05 | 1.11E-05 | 21.47% | 1.91% | 0.28% | 0.03% | 13.17% | 2.85% | 0.54% | 0.06% |
| | Yale-FinanceMath (Val) | Finance | 0.33 | 0.51 | 0.37 | 4.26E-05 | 2.23E-05 | 1.02E-05 | 24.11% | 3.08% | 0.50% | 0.05% | 18.73% | 5.35% | 1.23% | 0.15% |
| | Avg. | | 0.34 | 0.49 | 0.36 | 4.24E-05 | 2.16E-05 | 1.06E-05 | 24.29% | 3.44% | 0.73% | 0.11% | 17.04% | 5.00% | 1.39% | 0.27% |

Table 5 show that these suite of metrics serve as a comprehensive estimation of expertise which focus on different aspects of expertise, token level ($S_{spec}$, $R_{eff}$, $S_{iso}$), complexity metric ($\widehat{\mathcal{R}}_m(\mathbb{G}_{\theta*})$ -500, $\widehat{\mathcal{R}}_m(\mathbb{G}_{\theta*})$ -1000, $\widehat{\mathcal{R}}_m(\mathbb{G}_{\theta*})$ -2000) and sequence level ($\mathbb{E}_{NGR}^{(n)}$ -2, $\mathbb{E}_{NGR}^{(n)}$ -5, $\mathbb{E}_{NGR}^{(n)}$ -10, $\mathbb{E}_{NGR}^{(n)}$ -20, $\mathbb{G}_{G\text{-}NGR}^{(n)}$ -2, $\mathbb{G}_{G\text{-}NGR}^{(n)}$ -5, $\mathbb{G}_{G\text{-}NGR}^{(n)}$ -10, $\mathbb{G}_{G\text{-}NGR}^{(n)}$ -20).

*Table 5.* Expertise Results: Grouped by Partition

| Partition | Model | Domain | $S_{spec}$ | $R_{eff}$ | $S_{iso}$ | $\hat{\mathcal{R}}_m$-500 | $\hat{\mathcal{R}}_m$-1000 | $\hat{\mathcal{R}}_m$-2000 | $\mathbb{E}_{NGR}^{(n)}$-2 | $\mathbb{E}_{NGR}^{(n)}$-5 | $\mathbb{E}_{NGR}^{(n)}$-10 | $\mathbb{E}_{NGR}^{(n)}$-20 | $\mathbb{G}_{G\text{-}NGR}^{(n)}$-2 | $\mathbb{G}_{G\text{-}NGR}^{(n)}$-5 | $\mathbb{G}_{G\text{-}NGR}^{(n)}$-10 | $\mathbb{G}_{G\text{-}NGR}^{(n)}$-20 |
|---|---|---|---|---|---|---|---|---|---|---|---|---|---|---|---|---|
| AllenAI SciQ (Val) | DeepSeek-R1-0528 | Science | 0.24 | 0.47 | 0.31 | 3.45E-05 | 1.75E-05 | 8.54E-06 | 21.23% | 2.88% | 0.64% | 0.09% | 10.67% | 2.60% | 0.66% | 0.11% |
|  | DeepSeek-R1-0528 |  | 0.33 | 0.47 | 0.31 | 4.24E-05 | 2.20E-05 | 1.10E-05 | 25.35% | 3.72% | 0.77% | 0.10% | 13.92% | 3.60% | 0.83% | 0.12% |
|  | Qwen3-235B-Thinking |  | 0.70 | 0.26 | 0.45 | 4.34E-05 | 1.84E-05 | 9.65E-06 | 22.68% | 2.76% | 0.48% | 0.05% | 17.91% | 4.34% | 0.89% | 0.10% |
|  | Qwen3-30B-Instruct |  | 0.80 | 0.45 | 0.54 | 3.76E-05 | 2.15E-05 | 8.83E-06 | 27.47% | 5.70% | 1.46% | 0.24% | 29.00% | 11.01% | 3.78% | 0.81% |
|  | Qwen3-30B-Thinking |  | 0.79 | 0.45 | 0.43 | 4.09E-05 | 1.70E-05 | 9.54E-06 | 21.00% | 2.13% | 0.28% | 0.02% | 19.66% | 4.63% | 0.87% | 0.09% |
| BigBio MedQA (Dev) | DeepSeek-R1-0528 | Medical | 0.31 | 0.54 | 0.31 | 3.42E-05 | 1.68E-05 | 8.34E-06 | 19.43% | 2.45% | 0.48% | 0.06% | 13.08% | 3.18% | 0.78% | 0.11% |
|  | DeepSeek-R1-0528 |  | 0.47 | 0.59 | 0.39 | 3.98E-05 | 2.03E-05 | 9.98E-06 | 28.39% | 5.43% | 1.44% | 0.27% | 23.92% | 8.97% | 3.16% | 0.76% |
|  | Qwen3-235B-Thinking |  | 0.99 | 0.31 | 0.58 | 3.74E-05 | 1.99E-05 | 8.60E-06 | 28.50% | 4.87% | 1.10% | 0.17% | 36.77% | 15.15% | 5.50% | 1.39% |
|  | Qwen3-30B-Instruct |  | 0.90 | 0.50 | 0.57 | 3.74E-05 | 1.97E-05 | 9.63E-06 | 31.98% | 6.94% | 1.67% | 0.23% | 37.76% | 16.87% | 6.39% | 1.52% |
|  | Qwen3-30B-Thinking |  | 1.09 | 0.54 | 0.58 | 3.14E-05 | 1.41E-05 | 9.13E-06 | 27.82% | 4.70% | 0.97% | 0.12% | 37.28% | 15.05% | 5.59% | 1.73% |
| BigBio MedQA (Test) | DeepSeek-R1-0528 | Medical2 | 0.31 | 0.54 | 0.31 | 3.35E-05 | 1.70E-05 | 8.29E-06 | 19.37% | 2.41% | 0.47% | 0.06% | 13.02% | 3.11% | 0.75% | 0.11% |
|  | DeepSeek-R1-0528 |  | 0.47 | 0.59 | 0.39 | 4.18E-05 | 2.11E-05 | 1.01E-05 | 28.35% | 5.41% | 1.43% | 0.26% | 23.72% | 8.83% | 3.08% | 0.73% |
|  | Qwen3-235B-Thinking |  | 0.99 | 0.31 | 0.58 | 3.56E-05 | 1.75E-05 | 8.54E-06 | 28.51% | 4.87% | 1.11% | 0.17% | 36.71% | 15.10% | 5.47% | 1.39% |
|  | Qwen3-30B-Instruct |  | 0.91 | 0.50 | 0.57 | 2.93E-05 | 1.86E-05 | 1.00E-05 | 32.01% | 6.95% | 1.67% | 0.22% | 37.83% | 17.04% | 6.62% | 1.66% |
|  | Qwen3-30B-Thinking |  | 1.09 | 0.54 | 0.58 | 3.15E-05 | 1.76E-05 | 8.80E-06 | 27.84% | 4.72% | 0.99% | 0.13% | 37.23% | 15.07% | 5.65% | 1.76% |
| CAIS HLE | DeepSeek-R1-0528 | Knowledge | 0.10 | 0.28 | 0.22 | 3.70E-05 | 1.92E-05 | 8.96E-06 | 18.60% | 1.95% | 0.37% | 0.06% | 5.87% | 1.00% | 0.21% | 0.03% |
|  | DeepSeek-R1-0528 |  | 0.15 | 0.30 | 0.26 | 4.49E-05 | 2.37E-05 | 1.13E-05 | 24.20% | 3.61% | 0.83% | 0.15% | 10.99% | 2.94% | 0.84% | 0.18% |
|  | Qwen3-235B-Thinking |  | 0.49 | 0.24 | 0.37 | 4.85E-05 | 2.39E-05 | 1.28E-05 | 21.88% | 2.61% | 0.50% | 0.08% | 13.10% | 2.63% | 0.51% | 0.08% |
|  | Qwen3-30B-Instruct |  | 0.43 | 0.37 | 0.44 | 4.77E-05 | 2.38E-05 | 1.16E-05 | 22.01% | 2.57% | 0.57% | 0.17% | 14.40% | 3.59% | 0.80% | 0.12% |
|  | Qwen3-30B-Thinking |  | 0.47 | 0.39 | 0.38 | 4.36E-05 | 2.07E-05 | 1.04E-05 | 20.17% | 2.13% | 0.36% | 0.06% | 13.59% | 2.74% | 0.52% | 0.06% |
| LiveCodeBench (Test) | DeepSeek-R1-0528 | Code | 0.21 | 0.46 | 0.36 | 3.56E-05 | 1.77E-05 | 8.56E-06 | 21.72% | 2.85% | 0.60% | 0.10% | 12.26% | 3.36% | 0.93% | 0.20% |
|  | DeepSeek-R1-0528 |  | 0.30 | 0.46 | 0.34 | 4.22E-05 | 2.26E-05 | 1.13E-05 | 22.40% | 2.69% | 0.44% | 0.04% | 12.64% | 2.92% | 0.56% | 0.05% |
|  | Qwen3-235B-Thinking |  | 0.64 | 0.28 | 0.44 | 3.89E-05 | 1.92E-05 | 9.38E-06 | 21.59% | 2.62% | 0.48% | 0.07% | 21.11% | 5.82% | 1.50% | 0.26% |
|  | Qwen3-30B-Instruct |  | 0.44 | 0.39 | 0.54 | 4.49E-05 | 2.09E-05 | 9.79E-06 | 22.69% | 3.79% | 1.01% | 0.23% | 20.99% | 7.60% | 2.85% | 0.83% |
|  | Qwen3-30B-Thinking |  | 0.64 | 0.46 | 0.41 | 4.80E-05 | 2.06E-05 | 1.17E-05 | 19.13% | 1.99% | 0.34% | 0.04% | 19.77% | 4.92% | 1.16% | 0.18% |
| Nguha LegalBench | DeepSeek-R1-0528 | Legal | 0.31 | 0.56 | 0.39 | 3.40E-05 | 1.76E-05 | 8.21E-06 | 20.51% | 2.36% | 0.40% | 0.04% | 14.24% | 3.73% | 0.83% | 0.10% |
|  | DeepSeek-R1-0528 |  | 0.41 | 0.54 | 0.38 | 4.31E-05 | 2.01E-05 | 9.72E-06 | 24.59% | 3.34% | 0.64% | 0.08% | 18.72% | 5.57% | 1.43% | 0.24% |
|  | Qwen3-235B-Thinking |  | 0.94 | 0.27 | 0.56 | 3.43E-05 | 1.97E-05 | 9.05E-06 | 24.50% | 3.32% | 0.60% | 0.07% | 27.04% | 8.28% | 1.90% | 0.21% |
|  | Qwen3-30B-Instruct |  | 0.95 | 0.47 | 0.64 | 4.14E-05 | 1.85E-05 | 8.75E-06 | 25.69% | 5.31% | 1.50% | 0.31% | 30.67% | 11.92% | 4.09% | 0.86% |
|  | Qwen3-30B-Thinking |  | 0.98 | 0.48 | 0.53 | 3.45E-05 | 1.79E-05 | 7.26E-06 | 22.10% | 2.69% | 0.45% | 0.04% | 24.96% | 7.55% | 1.93% | 0.27% |
| Princeton SWE-bench (Test) | DeepSeek-R1-0528 | Code2 | 0.23 | 0.49 | 0.40 | 3.35E-05 | 1.69E-05 | 8.15E-06 | 22.07% | 2.62% | 0.48% | 0.06% | 14.81% | 3.90% | 0.89% | 0.12% |
|  | DeepSeek-R1-0528 |  | 0.23 | 0.42 | 0.39 | 4.50E-05 | 2.25E-05 | 1.11E-05 | 21.47% | 1.91% | 0.28% | 0.03% | 13.17% | 2.85% | 0.54% | 0.06% |
|  | Qwen3-235B-Thinking |  | 0.76 | 0.27 | 0.57 | 3.89E-05 | 1.90E-05 | 9.62E-06 | 21.41% | 1.71% | 0.19% | 0.02% | 20.57% | 4.92% | 0.86% | 0.08% |
|  | Qwen3-30B-Instruct |  | 0.63 | 0.44 | 0.62 | 3.74E-05 | 2.04E-05 | 1.05E-05 | 22.05% | 2.05% | 0.26% | 0.03% | 19.79% | 5.27% | 1.13% | 0.13% |
|  | Qwen3-30B-Thinking |  | 0.76 | 0.46 | 0.55 | 3.48E-05 | 2.04E-05 | 1.09E-05 | 20.24% | 1.57% | 0.19% | 0.03% | 23.85% | 6.51% | 1.42% | 0.17% |
| Yale-FinanceMath (Val) | DeepSeek-R1-0528 | Finance | 0.22 | 0.47 | 0.33 | 3.49E-05 | 1.71E-05 | 8.32E-06 | 18.82% | 2.00% | 0.39% | 0.07% | 13.28% | 3.21% | 0.75% | 0.13% |
|  | DeepSeek-R1-0528 |  | 0.33 | 0.51 | 0.37 | 4.26E-05 | 2.23E-05 | 1.02E-05 | 24.11% | 3.08% | 0.50% | 0.05% | 18.73% | 5.35% | 1.23% | 0.15% |
|  | Qwen3-235B-Thinking |  | 0.57 | 0.26 | 0.51 | 4.13E-05 | 1.93E-05 | 1.01E-05 | 24.53% | 3.58% | 0.68% | 0.09% | 26.29% | 8.46% | 2.27% | 0.39% |
|  | Qwen3-30B-Instruct |  | 0.57 | 0.41 | 0.58 | 4.60E-05 | 1.98E-05 | 1.05E-05 | 24.89% | 3.40% | 0.60% | 0.08% | 29.10% | 10.46% | 2.98% | 0.53% |
|  | Qwen3-30B-Thinking |  | 0.62 | 0.42 | 0.52 | 4.33E-05 | 2.15E-05 | 9.19E-06 | 22.22% | 2.66% | 0.43% | 0.04% | 24.32% | 6.96% | 1.52% | 0.17% |
| AIME 2025 | DeepSeek-R1-0528 | Math | 0.24 | 0.49 | 0.36 | 3.32E-05 | 1.71E-05 | 8.22E-06 | 18.69% | 1.87% | 0.30% | 0.04% | 13.93% | 3.12% | 0.63% | 0.08% |
|  | DeepSeek-R1-0528 |  | 0.35 | 0.49 | 0.42 | 3.97E-05 | 2.03E-05 | 1.07E-05 | 19.80% | 1.76% | 0.23% | 0.02% | 17.52% | 4.00% | 0.83% | 0.11% |
|  | Qwen3-235B-Thinking |  | 0.74 | 0.28 | 0.51 | 3.96E-05 | 1.87E-05 | 9.34E-06 | 19.26% | 1.73% | 0.25% | 0.04% | 21.20% | 5.34% | 1.13% | 0.16% |
|  | Qwen3-30B-Instruct |  | 0.63 | 0.44 | 0.56 | 3.71E-05 | 1.70E-05 | 1.07E-05 | 17.23% | 1.05% | 0.12% | 0.01% | 19.78% | 4.37% | 0.69% | 0.04% |
|  | Qwen3-30B-Thinking |  | 0.67 | 0.44 | 0.55 | 3.82E-05 | 1.92E-05 | 1.04E-05 | 17.18% | 1.31% | 0.15% | 0.01% | 20.02% | 5.14% | 1.00% | 0.09% |

## A.2. Specialty Radar Map of all metrics

Model-Centric Specialization Profiles. Figures 6 and 7 illustrate the functional specialization landscape across diverse architectures and domains. These radar maps provide a model-centric architectural fingerprint, serving as a qualitative counterpart to the quantitative aggregations in our previous tables. This diagnostic framework enables a comparative appraisal of expertise even in the absence of explicit pre-training data access. By mapping these specialized "tributaries," we can identify the most promising foundation models for targeted post-training interventions, ensuring that the selected base architecture possesses the requisite structural priors for specific downstream expertise.

Figures 6 and 7 illustrate the domain-specific specialty profiles of the evaluated models. We observe that DeepSeek-R1-0528 exhibits a remarkably balanced distribution across all evaluated domains, suggesting a versatile routing strategy that avoids over-specialization in any single niche. In contrast, GLM 4.6 demonstrates a pronounced functional alignment with the Medical domain, where its specialization metrics significantly outperform its own baseline in other areas.

The aggregate comparison in Figure 8 highlights a clear hierarchy in routing paradigms:

- Qwen3-MoE consistently occupies the outermost boundary of the radar map, representing the peak of structural specialization and isolation among the tested series.

- GLM 4.6 follows as the second most specialized, maintaining strong modularity while allowing for more synergy than Qwen.

- Conversely, DeepSeek-R1 displays the lowest magnitude across specialization-specific metrics.

This reinforces our earlier observation that DeepSeek prioritizes a generalized, high-entropy routing manifold over rigid domain boundaries, favoring synergetic expert coordination to handle diverse reasoning tasks.

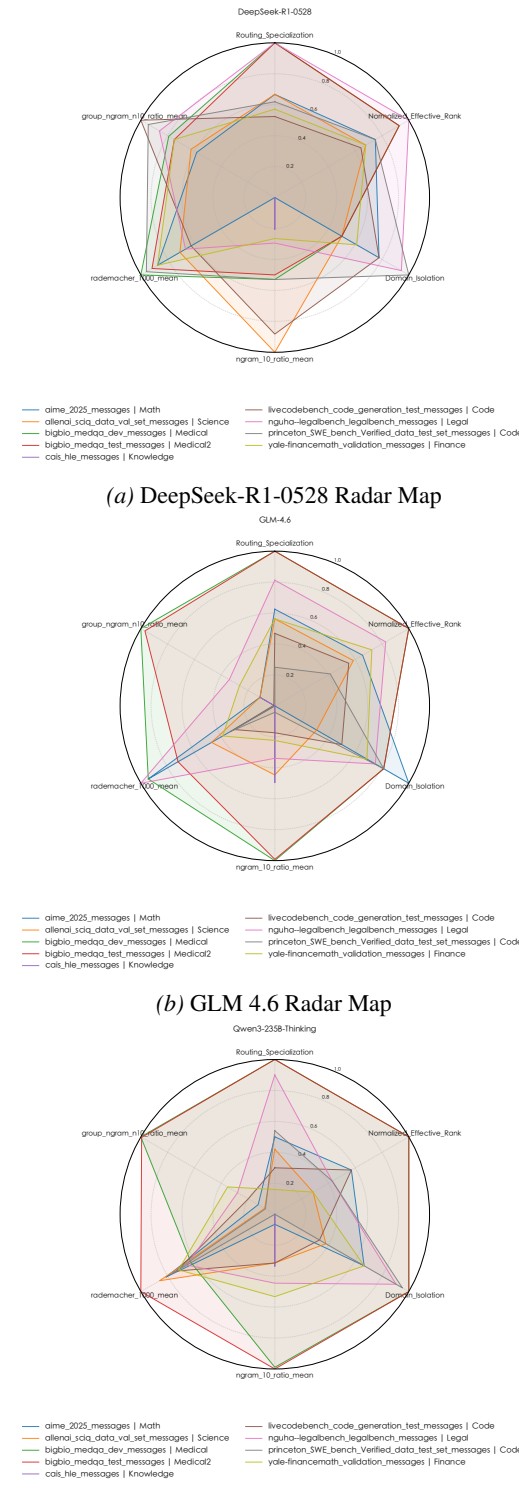

*(a)* DeepSeek-R1-0528 Radar Map

*(b)* GLM 4.6 Radar Map

*(c)* Qwen 235B Radar Map

*Figure 6.* **Domain-specific specialty fingerprints across LLM architectures.** The radar charts visualize relative expertise across diverse functional manifolds. Unlike static metric tables, these profiles reveal the structural orientation of each model, allowing for an informed selection of foundation models for domain-specific post-training specialization, regardless of the availability of original pre-training corpora.

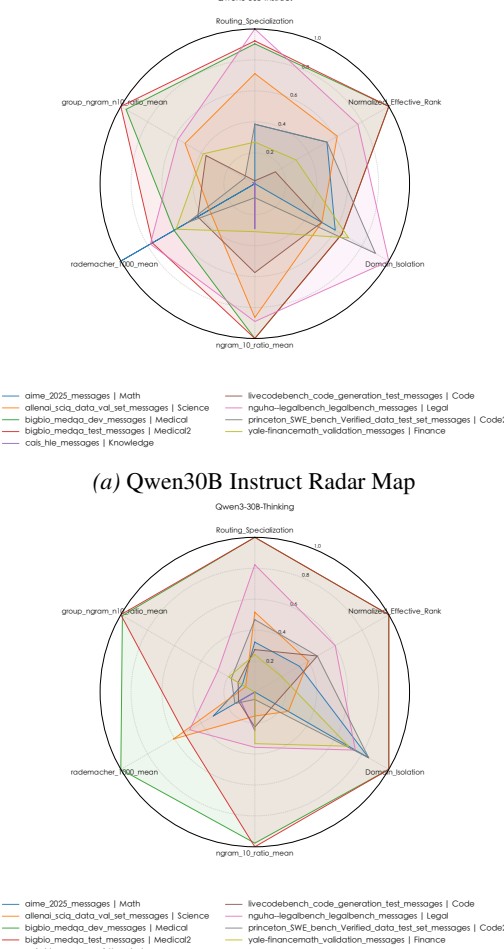

*(a)* Qwen30B Instruct Radar Map

*(b)* Qwen30B Thinking Radar Map

*Figure 7.* **Domain-specific specialty fingerprints across LLM architectures(continue).** The radar charts visualize relative expertise across diverse functional manifolds. Unlike static metric tables, these profiles reveal the structural orientation of each model, allowing for an informed selection of foundation models for domain-specific post-training specialization, regardless of the availability of original pre-training corpora

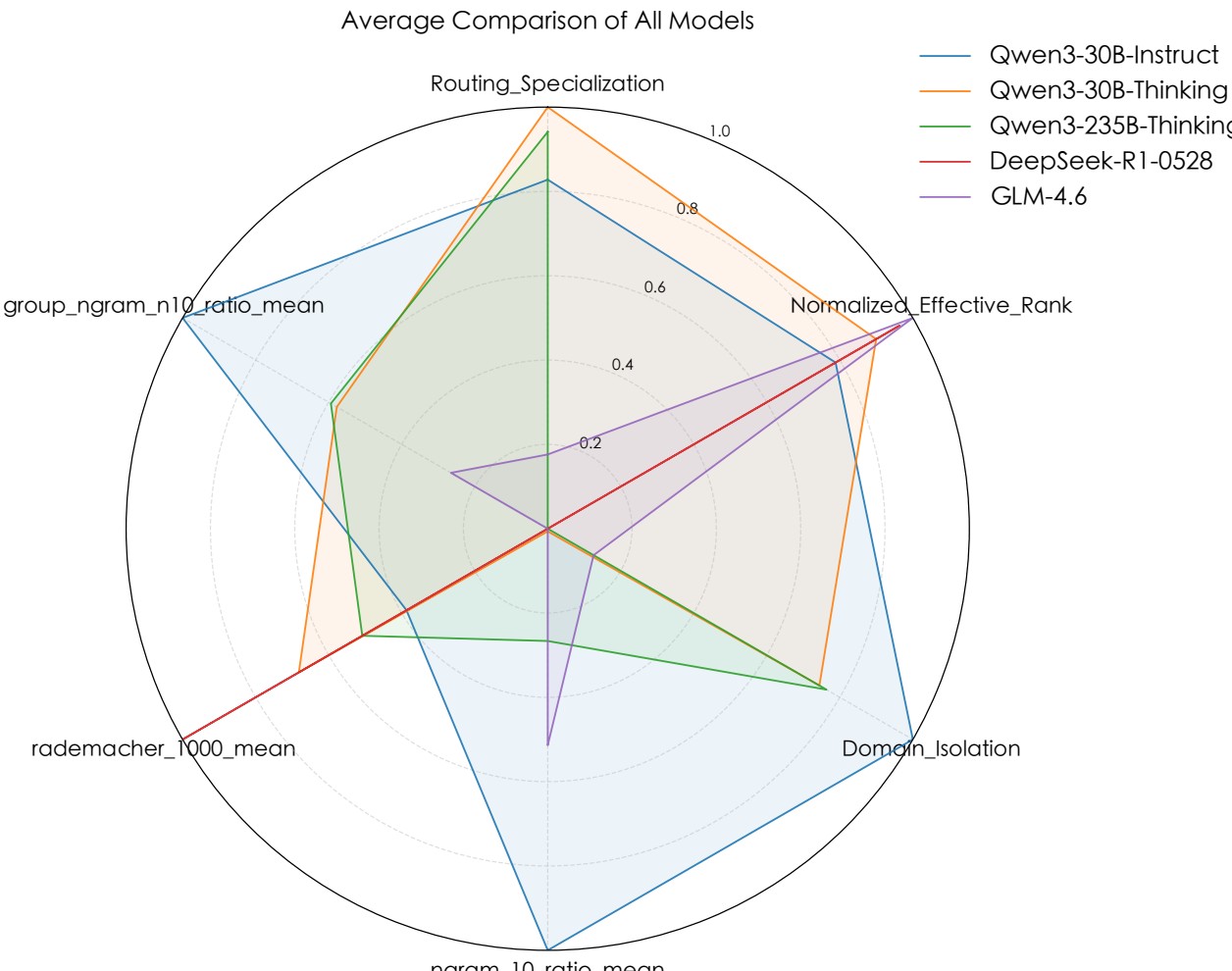

*Figure 8.* **Comparative analysis of expert specialization and routing paradigms.** a) Assessment of eleven metrics categorized into Specialization, Synergy, and Isolation across diverse domains. b) Contrast in routing behavior: DeepSeek-R1 exhibits lower Rademacher complexity, indicating high structural determinism; conversely, Qwen3-30B demonstrates superior N-gram expertise (temporal consistency), suggesting that its performance is critically dependent on sustained expert trajectories. c) Identification of two distinct paradigms: the Modular approach (Qwen: high specialization and isolation) with clear knowledge boundaries, and the Synergetic approach (DeepSeek/GLM: high effective rank) characterized by dynamic expert coordination.

## A.3. Activation Map of different Models in different domain

In Section 4, we show partial results, the full results is as Qwen3-30B-A3B-Thinking(Figure 9), Qwen3-30B-A3B-Instruct(Figure 10), Qwen3-235B-A22B-Thinking(Figure 11), GLM 4.6(Figure 12) and DeepSeek-R1-0528(Figure 13) While Section 4 highlights selected representative cases, the full suite of expert activation landscapescomprising Qwen3-30B-A3B-Thinking (Figures 9) and Qwen3-30B-A3B-Instruct (Figure 10), Qwen3-235B-A22B (Figure 11), GLM 4.6 (Figure 12), and DeepSeek-R1-0528 (Figure 13)is provided for a granular cross-model comparison.

These empirical results exhibit high internal consistency with our global radar map assessment. We observe distinct domain-specific activation signatures across all evaluated architectures, though the topology of these signatures varies significantly:

- The Qwen Series: Shows highly concentrated activation "hotspots" that are strictly partitioned by domain, reinforcing its status as a Modular routing paradigm.

- GLM & DeepSeek: Display more diffused yet structured activation patterns. As shown in the heatmaps, these models exhibit a synergetic distribution, where a core set of shared experts is dynamically augmented by domain-specialized expert clusters.

The clear divergence in expert-usage statistics between "Instruct" and "Thinking" versions of the same model (e.g., Figures 9 vs. 10) further validates our Sequence-Aware Bellman Rank hypothesis, indicating that increased reasoning depth (CoT) fundamentally reconfigures the routing manifold to favor more persistent, long-range expert trajectories.

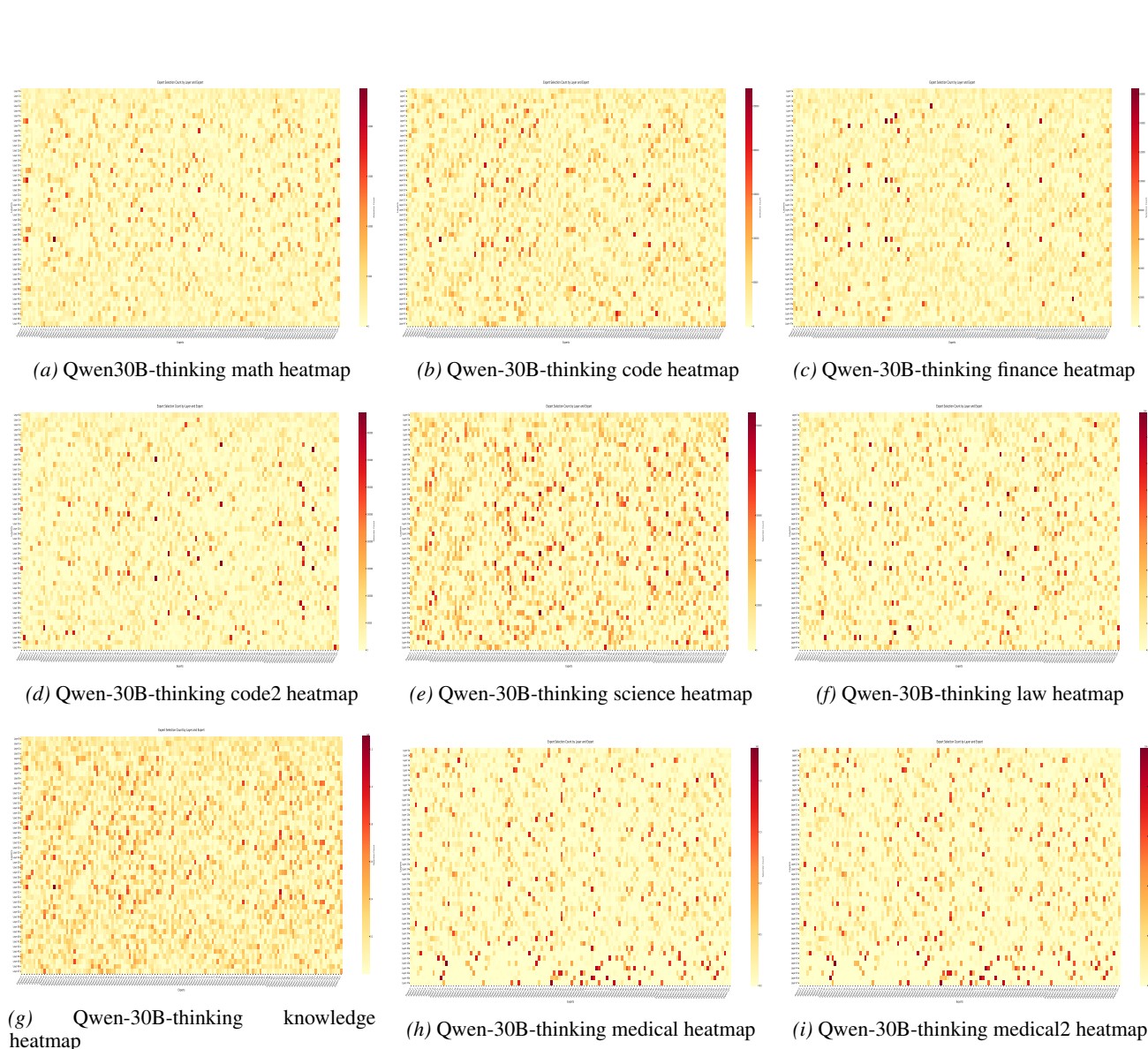

*(a)* Qwen30B-thinking math heatmap

*(b)* Qwen-30B-thinking code heatmap

*(c)* Qwen-30B-thinking finance heatmap

*(d)* Qwen-30B-thinking code2 heatmap

*(e)* Qwen-30B-thinking science heatmap

*(f)* Qwen-30B-thinking law heatmap

*(g)* Qwen-30B-thinking knowledge heatmap

*(h)* Qwen-30B-thinking medical heatmap

*(i)* Qwen-30B-thinking medical2 heatmap

*Figure 9.* **Qwen3-30B-A3B-Thinking model activation.** a) Heterogeneous Expert Concentration: The model exhibits domain-specific activation densities; scientific domains are characterized by dense, clustered expert usage, whereas code and math domains demonstrate comparative sparsity in expert selection. b) Intradomain Variability: Significant fluctuations in activation intensity are observed even within the same domain; for example, different coding tasks trigger distinct expert sub-manifolds, reflecting the model's granular adaptability to diverse problem complexities.

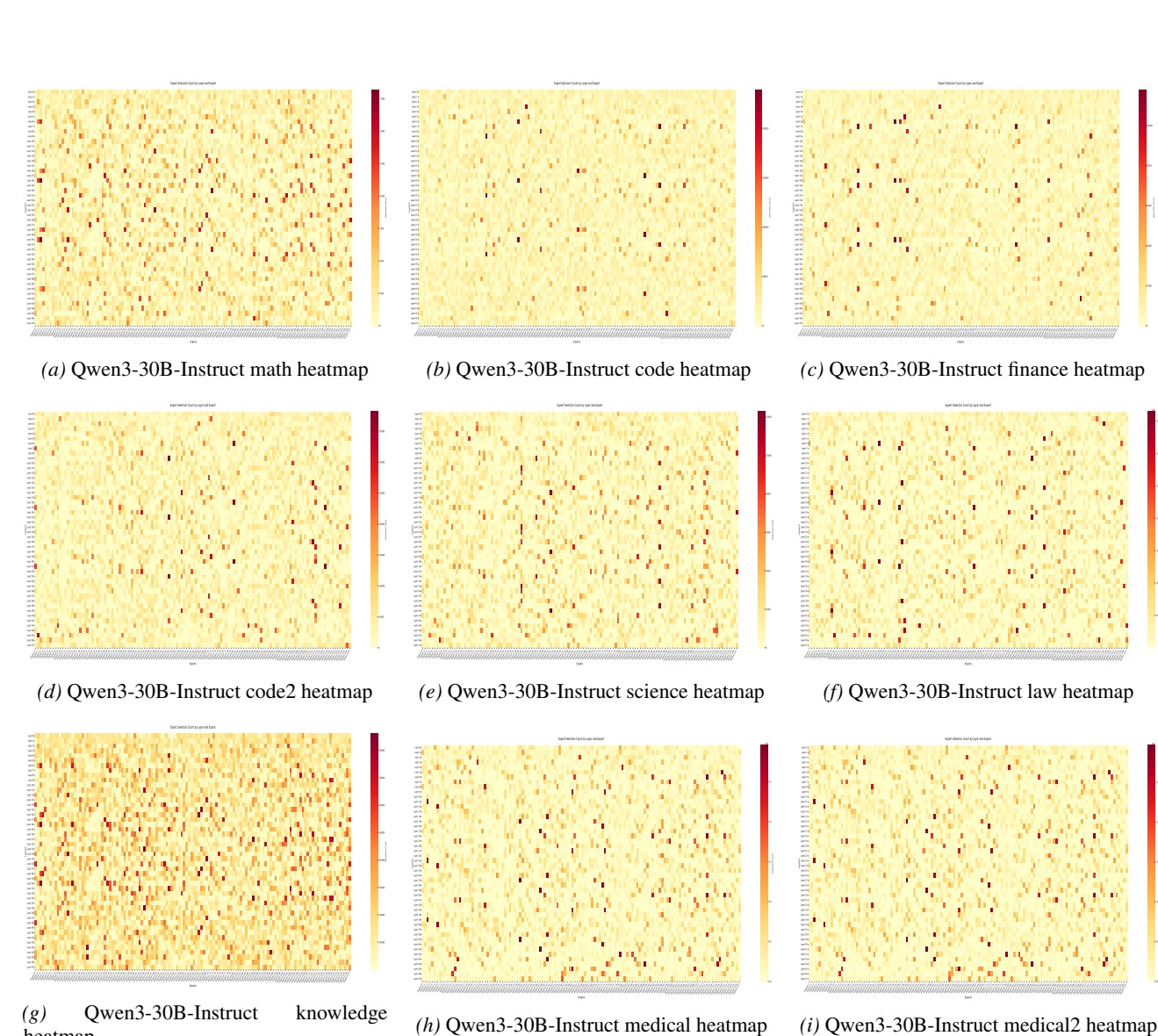

*(a)* Qwen3-30B-Instruct math heatmap   *(b)* Qwen3-30B-Instruct code heatmap   *(c)* Qwen3-30B-Instruct finance heatmap

*(d)* Qwen3-30B-Instruct code2 heatmap   *(e)* Qwen3-30B-Instruct science heatmap   *(f)* Qwen3-30B-Instruct law heatmap

*(g)* Qwen3-30B-Instruct knowledge heatmap   *(h)* Qwen3-30B-Instruct medical heatmap   *(i)* Qwen3-30B-Instruct medical2 heatmap

*Figure 10.* **Qwen3-30B-A3B-Instruct model activation.** a) Activation Gradients: Intensive expert recruitment is concentrated in knowledge and math domains, with pronounced sparsity elsewhere. b) Intra-domain Variance: Comparison between Code and Code2 reveals a shift in expert preference, suggesting that task-specific logic (e.g., script vs. algorithmic) triggers different expert sub-manifolds.

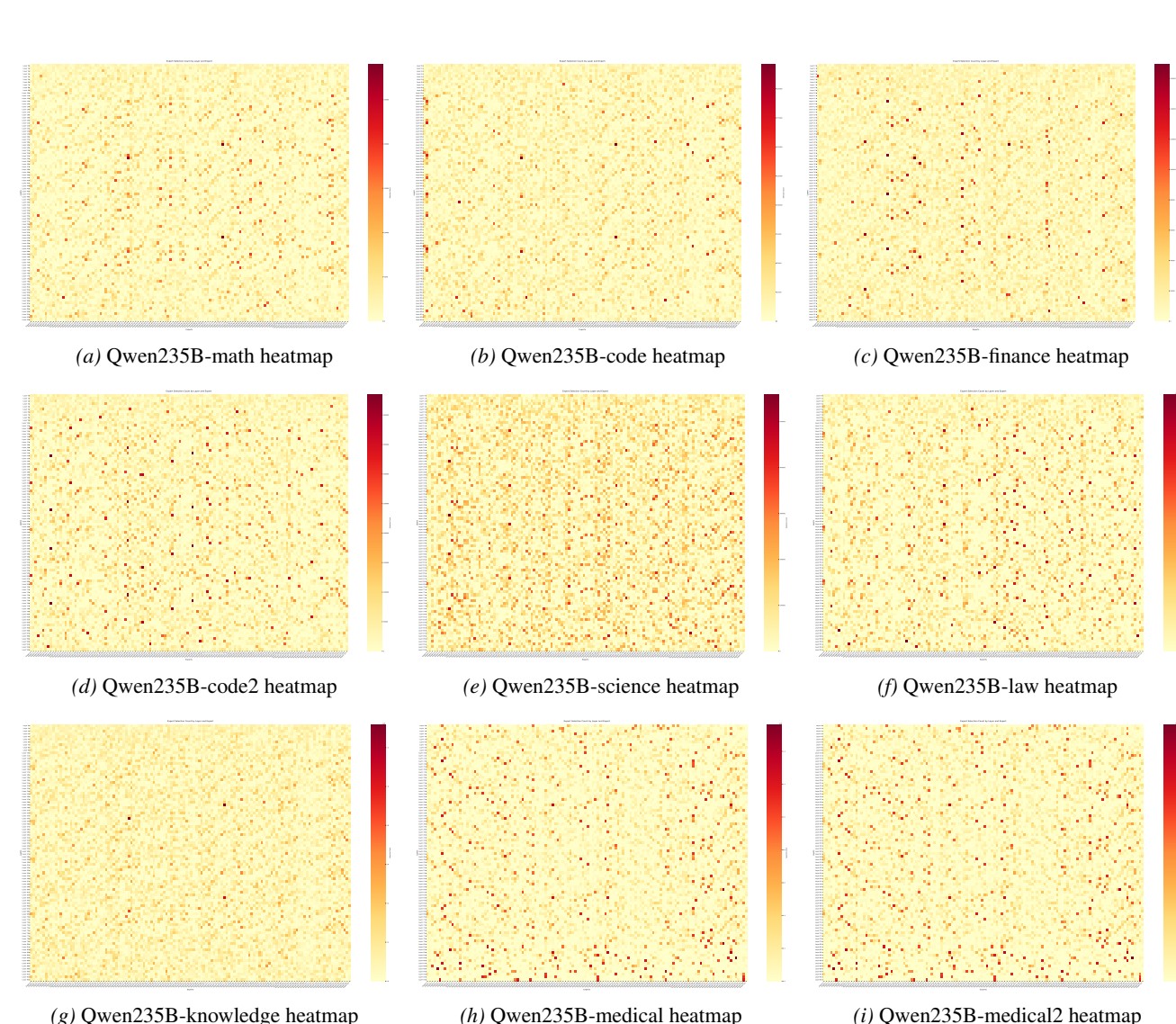

*(a)* Qwen235B-math heatmap  *(b)* Qwen235B-code heatmap  *(c)* Qwen235B-finance heatmap

*(d)* Qwen235B-code2 heatmap  *(e)* Qwen235B-science heatmap  *(f)* Qwen235B-law heatmap

*(g)* Qwen235B-knowledge heatmap  *(h)* Qwen235B-medical heatmap  *(i)* Qwen235B-medical2 heatmap

*Figure 11.* **Qwen3-235B-A22B-Thinking model activation.** a) Divergent Activation Densities: High-density expert recruitment is observed in science and law, while the knowledge domain exhibits a near-uniform distribution across the manifold, indicating a lack of dominant expert "hotspots." b) Sub-task Bifurcation: Expert selection for Code and Code 2 reveals a clear trajectory shift, reflecting task-specific routing adaptability in high-parameter MoE architectures.

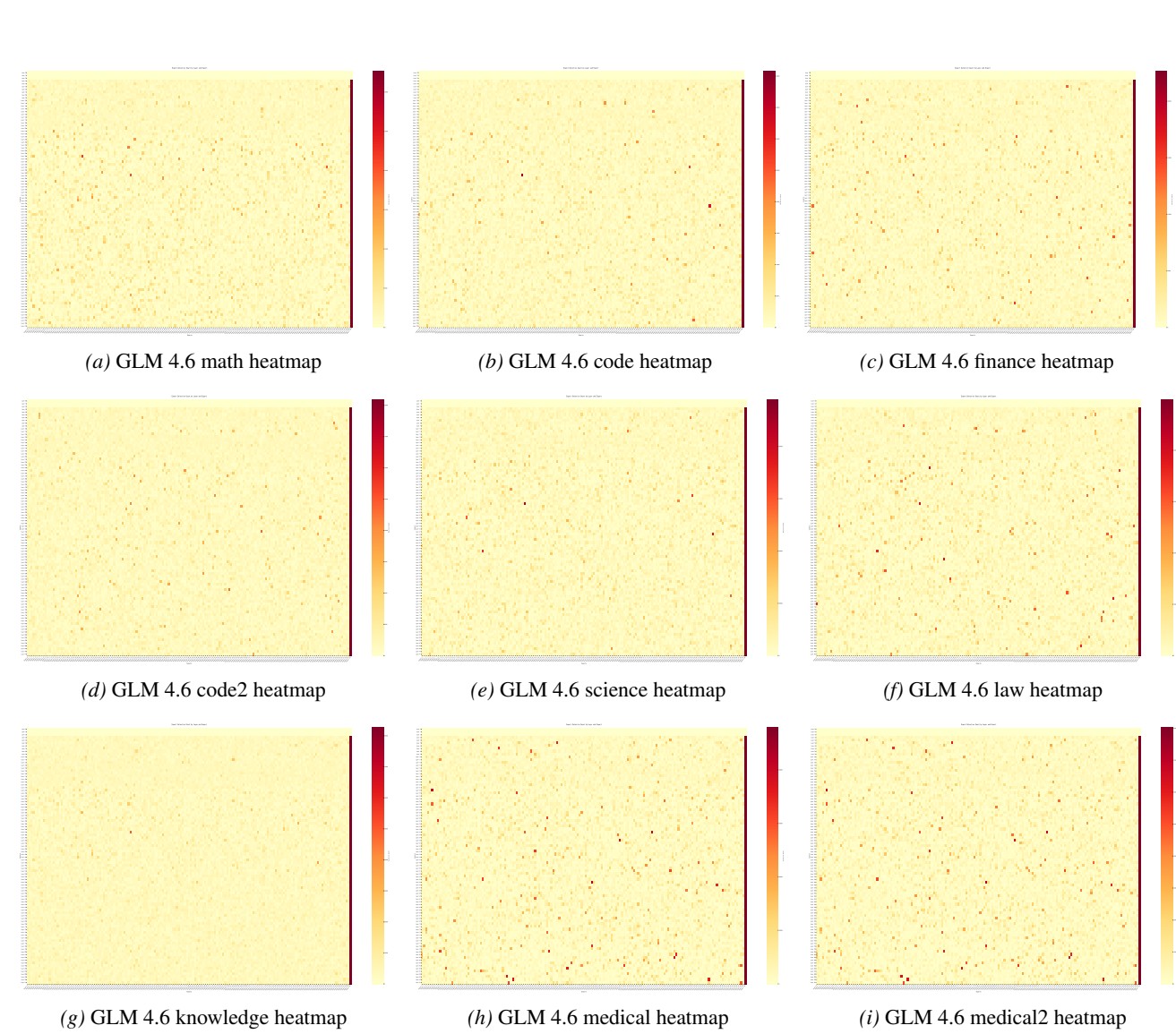

*(a)* GLM 4.6 math heatmap

*(b)* GLM 4.6 code heatmap

*(c)* GLM 4.6 finance heatmap

*(d)* GLM 4.6 code2 heatmap

*(e)* GLM 4.6 science heatmap

*(f)* GLM 4.6 law heatmap

*(g)* GLM 4.6 knowledge heatmap

*(h)* GLM 4.6 medical heatmap

*(i)* GLM 4.6 medical2 heatmap

*Figure 12.* **GLM 4.6 model activation.** a) Distributed Synergy: The model shows near-uniform expert activation across diverse tasks, suggesting a high-entropy routing approach with minimal cross-domain divergence. b) Selective Expertise: A localized increase in expert recruitment intensity is observed in the medical domain, representing a distinct functional preference within an otherwise homogeneous routing manifold.

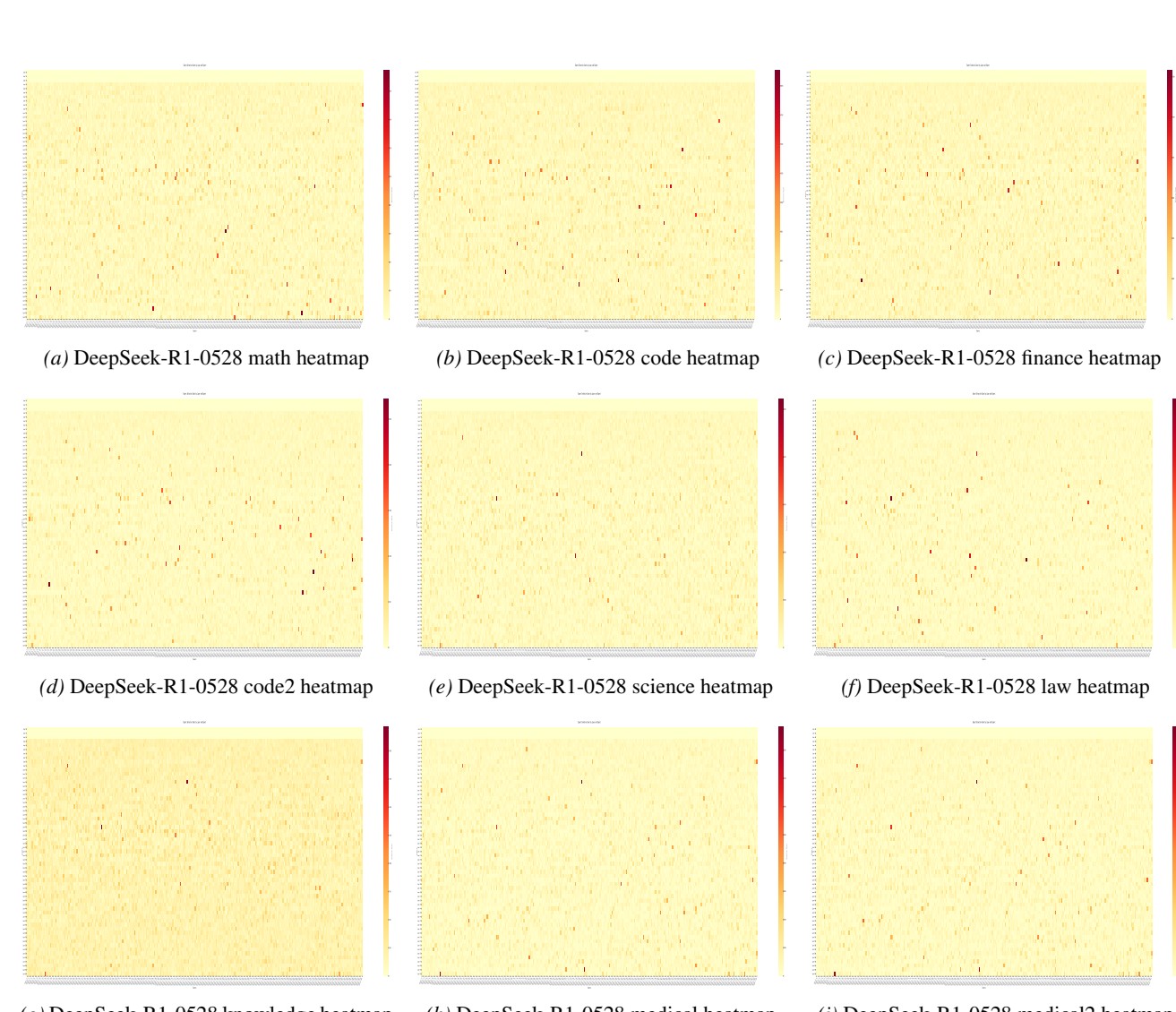

*(a)* DeepSeek-R1-0528 math heatmap  *(b)* DeepSeek-R1-0528 code heatmap  *(c)* DeepSeek-R1-0528 finance heatmap

*(d)* DeepSeek-R1-0528 code2 heatmap  *(e)* DeepSeek-R1-0528 science heatmap  *(f)* DeepSeek-R1-0528 law heatmap

*(g)* DeepSeek-R1-0528 knowledge heatmap  *(h)* DeepSeek-R1-0528 medical heatmap  *(i)* DeepSeek-R1-0528 medical2 heatmap

*Figure 13.* **DeepSeek-R1-0528 model activation.** a) Distributed Load Balancing: The model demonstrates near-equivalent expert utilization across diverse tasks, indicating a high-entropy routing policy that maintains a balanced expert load. b) Task-Specific Intensity: Pronounced activation "hotspots" appear specifically in the code domain, highlighting a distinct functional preference for programming and logic-heavy contexts within an otherwise homogeneous manifold.

# B. MoE Specialization and Performance

A specified expert will give the insight of specialized fields, but if the specialty reduce the experts performance, this will make the system collapse thus we analyze the specialty along with performance on some domain benchmarks.

## B.1. Theoretical Proof of Error Bound using NGR

First, Zhao et al. have proved using mathematical computation. We apply the findings to Qwen3, GLM and DeepSeek.

**Lemma B.1** (Generalization Bound for Sparse MoE)**.** *Let the MoE model be $C$-Lipschitz and the MLP layers be defined by a neural network function class with a fixed structure $S$. Given input $\|x^{(i)}\|_2 \leq c$, and weight matrices constrained by spectral norms $\|W_i\|_{op} \leq K_i$, the generalization error upper bound is:*

$$O\left(4C\frac{c}{\sqrt{m}} \cdot \left(\prod_{i=1}^{r} K_i\right) \cdot \left(\sum_{i=1}^{r} \frac{b_i^{2/3}}{K_i^{2/3}}\right)^{3/2} + 2\sqrt{\frac{2kdp^2\left(1 + \log\left(\frac{T}{k}\right)\right) + dp^2\log(2m) + \log(\frac{4}{\delta})}{2m}}\right). \qquad (11)$$

Applying Lemma B.1 to architectures like Qwen3-MoE, GLM and DeepSeek-MoE, we observe that Top-$K$ selection acts as a regularizer. The term $\log(\frac{T}{k})$ explicitly shows that as the total experts $T$ grow relative to selected $k$, the model requires a logarithmic increase in samples to maintain the generalization gap.

## B.2. Benchmark Performance vs. Specialty

We investigate whether structural expertise correlates with the model's objective performance across diverse reasoning tasks. Table 6 presents benchmark results compiled from official technical reports. While certain benchmarks are domain-specific and lack exhaustive cross-model testing, the current data highlights a significant trend. Notably, DeepSeek-R1-0528 achieves highly competitive results, particularly in specialized domains such as coding, advanced mathematics, and scientific reasoning. These findings support our hypothesis: a high degree of architectural specialization (Expertise) directly translates to superior performance in complex, high-entropy tasks.

*Table 6.* Comparison of Qwen3, GLM 4.6 and DeepSeek-R1 benchmarks.

| ID | Dataset | Q3-32B | Q3-MoE | Q3-235B | GLM | DS-R1 | Notes |
|---|---|---|---|---|---|---|---|
| aime | AIME 2025 | 72.9% | 70.9% | 81.5% | 93.9% | 70.0% | Advanced math reasoning. |
| CAIS HLE | HLE | 8.3% | 8% | 18.2% | 17.2% | 17.7% | Extremely difficult 2025 benchmark. |
| live | LCB | 68.4% | 62.6% | 74.1% | **82.8%** | 68.7% | |
| swe | SWE-bench | – | – | – | **68.0%** | 57.6% | Verified set. |

While Qwen3-30B exhibits competitive specialization, its absolute performance is constrained by its active parameter count. Consequently, we focus our correlation analysis on Qwen3-235B, DeepSeek-R1-0528, and DeepSeek-R1, which possess comparable effective inference capacities ( 22B-32B active parameters).

Figure 14 demonstrates a robust positive correlation between architectural specialty and task performance, suggesting that superior reasoning capabilities are inherently tied to domain-specific expert grounding. A granular analysis in Figure 15 reveals nuanced relationships between individual metrics and accuracy. Notably, Rademacher Complexity serves as a strong negative indicator ($r = -0.3411(500), -0.3774(1000), -0.2676(2000)$), reinforcing the premise that lower routing complexity (higher determinism) favors model proficiency.

**The Stability Paradox.** Intriguingly, while most sequence-level metrics align positively with performance, the N-gram ratios for longer windows ($n = 10, 20$) exhibit a surprising inverse correlation ($r = -0.5675$ and $-0.6545$, respectively). This divergence suggests two underlying mechanisms:

- **Hierarchical Routing Efficiency:** Top-performing models like GLM-4.6 and DeepSeek-R1 leverage a hierarchical "Group-Expert" strategy. Our results show that while expert-level persistence may fluctuate, the Group N-gram ratio maintains a strong positive correlation (0.6663 for $n = 2$ and 0.6201 for $n = 5$). This implies that the model's "expertise" is anchored at the group (domain) level, allowing for dynamic expert switching within a stable functional cluster.

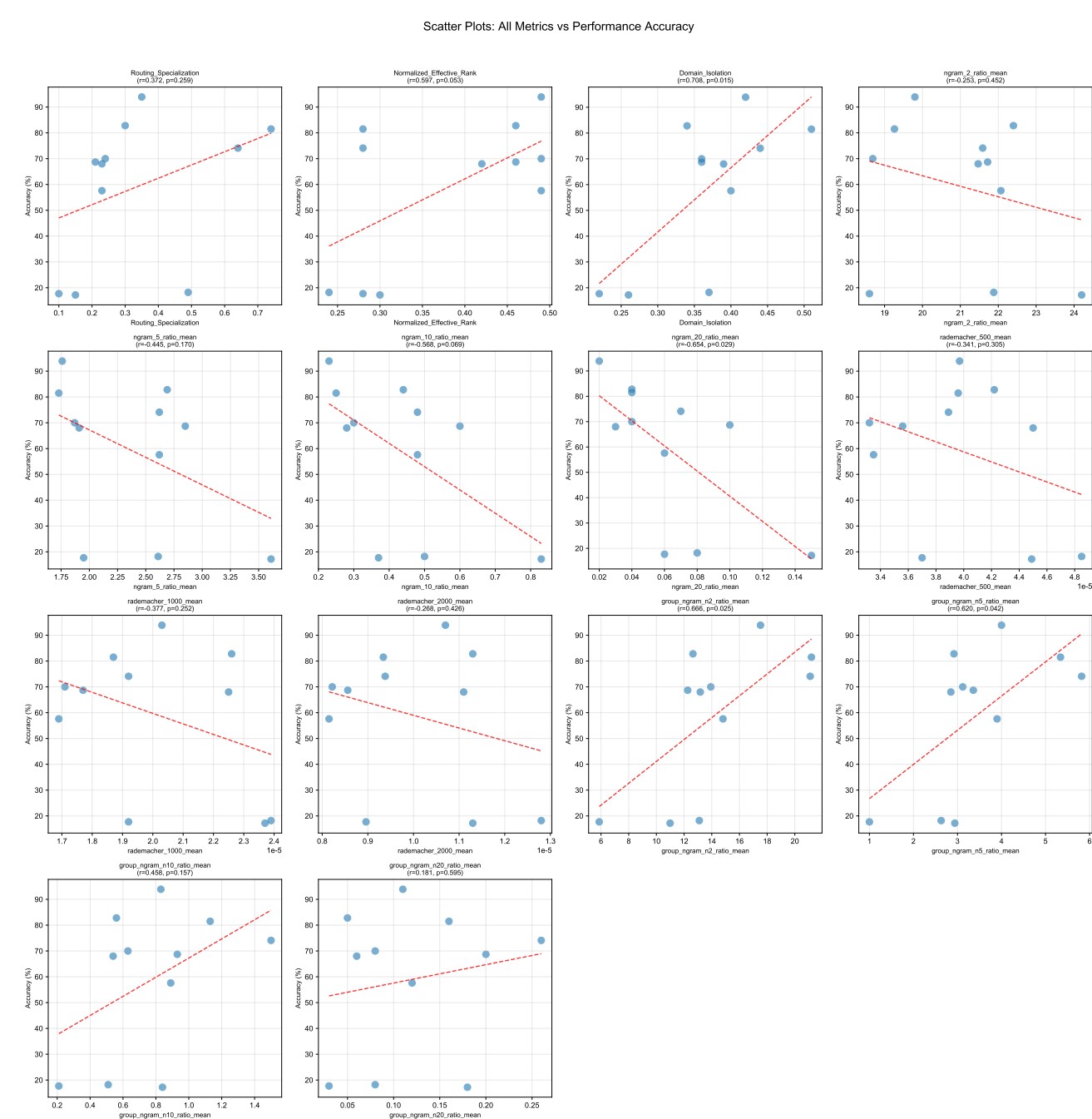

*Figure 14.* **Correlation of Performance with each metric.** a) Positive Predictors: Core specialization metricsincluding Routing Specialization ($S_{spec}$), Normalized Effective Rank ($R_{eff}$), Domain Isolation ($S_{iso}$), and long-range temporal stability (N-gram ratio-10 and Group N-gram ratio-10)exhibit a strong positive correlation with model accuracy. This alignment suggests that macroscopic domain-anchoring and sustained expert trajectories are primary drivers of reasoning proficiency. b) Inverse Indicators: Rademacher Complexity and short-range expert transitions (N-gram-2, N-gram-5) show a negative correlation with performance. This inverse relationship indicates that while micro-level expert switching is inherent to the MoE architecture, excessive routing stochasticity and lack of long-range expert commitment (high Rademacher complexity) can be detrimental to specialized problem-solving.

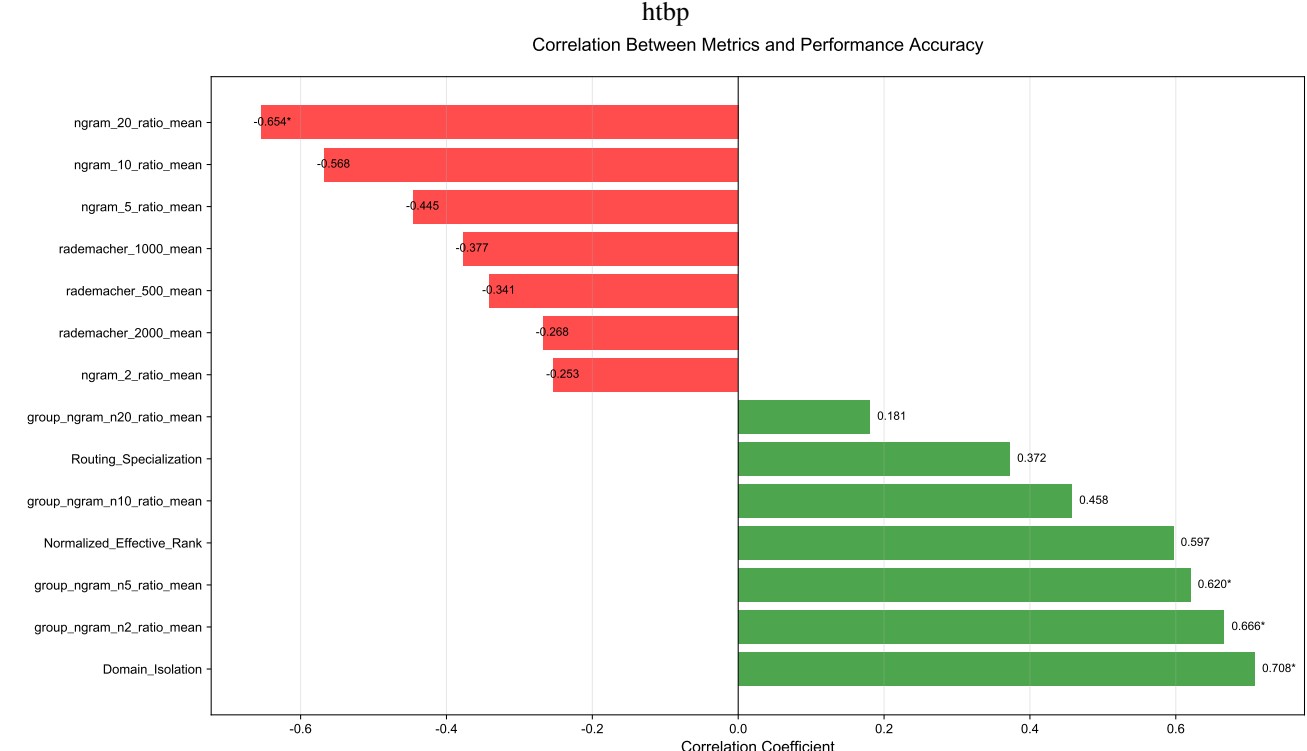

*Figure 15.* **Correlation of Performance with each individual metric.** a) Inverse Indicators (Red Bars): Metrics such as Rademacher Complexity and short-range expert transitions ($n = 2, 5$) are negatively correlated with accuracy. This suggests that excessive routing stochasticity and micro-level "expert jitter" act as structural noise that compromises reasoning stability. b) Positive Predictors (Green Bars): Core metricsincluding Routing Specialization ($S_{spec}$), Normalized Effective Rank ($R_{eff}$), Domain Isolation ($S_{iso}$), and long-range temporal consistency ($n = 10$)show a strong positive correlation with performance. These results demonstrate that a specialized, modular, and temporally persistent expert manifold is a fundamental driver of superior model proficiency.

- **Impact of Training Constraints:** The prevalence of forced Load-Balancing losses during training across these SOTA models likely induces divergent expert utilization. This pressure to distribute tokens across the expert manifold may break long-range expert-level consistency ($n \geq 10$) while preserving macro-level domain specialization.

# C. N-gram Expertise–Theretical Analysis

## C.1. MoE as Bellman Optimization System

### C.1.1. WHY VIEW MOE AS BELLMAN OPTIMIZATION

**Action Space** ($\mathcal{A}$)   The decision-maker (router) at each layer $l$ for each sequence index $t_i$ selects a subset of experts $K_{l,t_i} \subset \{1, \ldots, E\}$. For a Top-$k$ routing policy, the action at a specific temporal-layer coordinate $(l, t_i)$ is:

$$a_{l,t_i} = K_{l,t_i} \in K \subseteq [E] : |K| = k \tag{12}$$

**Transition Dynamics**   The transition occurs along the depth axis (layer-wise) for each token in the sequence. Let $\mathbb{G} : \mathcal{S} \to \mathbb{R}^E$ be the gating function and $\mathbb{E}_e : \mathcal{S} \to \mathcal{S}$ be the $e$-th expert function. The state evolution for a token $t_i$ from layer $l$ to $l+1$ is defined as:

$$s_{l+1,t_i} = \text{Trans}(s_{l,t_i}, K_{l,t_i}) = \sum_{e \in K_{l,t_i}} \mathbb{G}e(sl, t_i) \cdot \mathbb{E}e(sl, t_i) \tag{13}$$

This formulation allows the Bellman system to track the "history" of a token's representation as it traverses the expert manifold.

**Reward and Policy**   The policy $\pi$ dictates the expert selection based on the local hidden state: $\pi(s_{l,t_i}) = \text{arg-TopK}(\mathbb{G}(s_{l,t_i}))$. We define the reward $r(s_{l,t_i}, K_{l,t_i})$ as the contribution to the negative differential loss. Crucially, for our specialized RL training plan, we define the **Expertise-Augmented Reward**:

$$r_{aug}(s_{l,t_i}, K_{l,t_i}) = r_{task} + \lambda \cdot \mathbb{I}(\text{consistency across } t_{i-n:i}) \tag{14}$$

where the second term explicitly rewards the temporal consistency of expert selection across the $n$-gram window.A.1.3 Bellman Operators and Error Matrix

**Sequence-Aware Bellman Operator**   For a value function $V : \mathcal{S} \to \mathbb{R}$, the Bellman operator $\mathcal{T}^\pi$ now operates on the sequence-indexed state:

$$(\mathcal{T}^\pi V)(s_{l,t_i}) = r(s_{l,t_i}, \pi(s_{l,t_i})) + \gamma V(s_{l+1,t_i}) \tag{15}$$

where $\gamma$ is the discount factor representing the decay of influence across layer depth.

**Bellman Error Matrix** ($\mathbb{M}$)   The error matrix $\mathbb{M}$ is constructed over the set of observed tokens across the corpus and the set of routing policies $\Pi$:

$$\mathbb{M}(t_i, l), \pi\theta = \mathcal{E}(s_{l,t_i}, \pi_\theta) = (\mathcal{T}^{\pi_\theta} V)(s_{l,t_i}) - V(s_{l,t_i}) \tag{16}$$

The Bellman Rank $\kappa = \text{rank}_\epsilon(\mathbb{M})$ thus measures the complexity of the routing task across both the depth of the model and the temporal structure of the language.

### C.1.2. PROOF OF LOW-RANK STRUCTURE

**Theorem C.1.** *If the gating function $\mathbb{G}$ and expert functions $\mathbb{E}_e$ are Lipschitz continuous, then the Bellman Error Matrix $\mathbb{M}$ possesses a low numerical rank $\kappa$.*

*Proof.* Consider the mapping $(s, \theta) \mapsto \mathcal{E}(s, \pi_\theta)$.

1. **Lipschitz Continuity:** By assumption, $\mathbb{G}$ is $C$-Lipschitz. Since the transition dynamics and rewards are compositions of $\mathbb{G}$ and $\mathbb{E}_e$, the function $\mathcal{E}$ is also Lipschitz continuous.

2. **Function Approximation:** For any $\epsilon > 0$, the space of Lipschitz functions over a compact domain can be $\epsilon$-approximated by a finite set of basis functions $\{\psi_i\}_{i=1}^N$.

3. **Matrix Factorization:** Each entry $\mathbb{M}_{s,\pi_\theta}$ can be approximated as $\sum_{i=1}^\kappa u_i(s)v_i(\theta)$. The number of required components $\kappa$ scales with the covering number of the function class, which is logarithmic relative to the precision $\epsilon$.

Thus, $\kappa \leq O(d \log \frac{CR}{\epsilon})$, where $d$ is the intrinsic dimension of the representation space. $\square$

### C.1.3. WHY MAKE MOE A BELLMAN OPTIMIZATION SYSTEM

While this paper focuses on the appraisal of existing specialization, the theoretical framework provided by the Bellman Rank and NGR serves as the objective function for a future Reinforcement Learning of Expert Specialization training stage.

**Planned Reward Formulation**  We propose a composite reward signal $R_{\text{spec}}$ to be used in post-training to incentivize experts to branch into specialized "tributaries" rather than collapsing into the shared "mainstream" knowledge:

$$R_{\text{spec}} = \alpha \cdot \mathbb{E}_{\text{NGR}}^{(n)} - \beta \cdot \widehat{\mathcal{R}}m(\mathbb{G}_{\theta^*}) \tag{17}$$

where $\alpha$ rewards temporal consistency (expertise) and $\beta$ penalizes routing noise (complexity).

**Implementation Note**  This RL-based training paradigm is currently in the design phase. The metrics introduced in this work (RS, NER, DI, and NGR) provide the necessary diagnostic tools to verify if such a reward system successfully induces functional isolation without compromising the model's foundational reasoning capabilities.

### C.2. MoE with Mutual Information

*Proof.* The proof proceeds by bounding the probability of the "constant path" event using information-theoretic inequalities.

### C.2.1. MUTUAL INFORMATION DECOMPOSITION

By the definition of Mutual Information:

$$I(x_{1:n}; p_{1:n}) = H(p_{1:n}) - H(p_{1:n}|x_{1:n}) \tag{18}$$

In a learned MoE, the gating policy $\mathbb{G}$ minimizes the conditional entropy $H(p|x)$ to ensure deterministic expert specialization. Thus, as $H(p_{1:n}|x_{1:n}) \to 0$, the mutual information $I_n(x; p)$ is lower-bounded by the entropy of the routing distribution $H(p_{1:n})$.

### C.2.2. ENTROPIC CHAIN RULE AND LOCAL CONSISTENCY

We expand the joint entropy of the routing path using the chain rule:

$$H(p_{1:n}) = \sum_{t=1}^{n} H(p_t|p_{1:t-1}) \tag{19}$$

High $\mathbb{E}_{\text{NGR}}^{(n)}$ implies that the transition probability $P(p_t = p_{t-1}|p_{t-1})$ is high. Let $\sigma$ denote the event of a consistent path $(p_1 = p_2 = \cdots = p_n)$. In the manifold of all possible paths $\Pi^{(n)}$, the N-gram Expertise is the empirical measure of this specific subset:

$$\mathbb{E}_{\text{NGR}}^{(n)} = P(\sigma) = \frac{\text{Card}(\omega \in \Omega^{(n)} : p_1 = \cdots = p_n)}{\text{Card}(\Omega^{(n)})} \tag{20}$$

### C.2.3. CONCENTRATION INEQUALITY

Using the relationship between entropy and the probability of the most likely sequence (Fano's inequality variant), we observe that $H(p_{1:n})$ is maximized when the distribution is uniform. Conversely, a high $I_n(x; p)$ signifies that the input $x$ significantly reduces the effective search space of paths.If $I_n(x; p)$ is large, the router must "lock" onto specific high-probability trajectories. Given that natural language inputs $x_{1:n}$ exhibit high temporal redundancy (low $H(x_t|x_{t-1})$), a router maximizing mutual information will mirror this redundancy in the routing space, resulting in:

$$H(p_{1:n}) \leq H_{\max} - I_n(x; p) \tag{21}$$

### C.2.4. CONCLUSION

Applying the exponential map to the entropy bound and normalizing by the expert manifold volume $E^{n-1}$, we obtain:

$$\mathbb{E}\mathrm{NGR}^{(n)} \propto \exp(-H(p1:n)) \geq \frac{1}{E^{n-1}} \exp\left(\frac{I_n(x;p) - H_{\max}}{n}\right) \tag{22}$$

This confirms that the N-gram Expertise is a lower bound on the information-theoretic coupling between inputs and routing decisions. $\square$

Refined Logic Notes: $H(p|x) \to 0$: This assumes the router has "learned" its specialty. If the router is random, NGR is low and MI is low. Normalization: The $E^{n-1}$ factor accounts for the fact that there are $E^n$ total paths, but only $E$ possible "constant" paths (e.g., $1-1-1$, $2-2-2$, etc.). The Log-Exp link: This bridges the gap between the bits of information (MI) and the physical count of repeated paths (NGR).

# D. Data analysis of interlayer moe distribution and expertise – focus on 1st, middle and last layers

## D.1. Layer-wise Routing Specialization Trajectories

Figure 17 visualizes the evolution of expert focus with a unified Y-axis scale. Three key topological patterns emerge:

- **Universal Initialization:** All models start with near-zero specialization, confirming a universal "Shared Stem" for domain-agnostic feature extraction in shallow layers.

- **Magnitude Disparity:** A structural divergence is evidentQwen models adopt a "Specialist" strategy (maintaining $S_{spec} > 1.0$), while DeepSeek and GLM function as "Generalists" ($S_{spec} < 0.6$).

- **Terminal Bifurcation:** The "Low-High-Low" convergence hypothesis holds for standard Instruct models. However, CoT-finetuned models (Fig. 17 b/c) defy this trend, exhibiting a **"Terminal Surge"** where deep layers are repurposed for highly isolated, domain-specific reasoning.

## D.2. Cross-Domain Similarity in Critical Layers

We visualize heatmaps of expert choice similarity between domains at the first, middle, and last layers to track how knowledge sharing evolves.

The heatmaps(Figure 16) reveal two distinct routing strategies. Qwen-235B exhibit an "High-Low-High" pattern for most domains: tasks share the input layer, separate in the middle for specialized reasoning, and re-integrate at the output layer to share a common representation. A notable exception is Qwen's Medical domain, which remains strictly isolated (similarity about 0.2) from the very first layer. In contrast, DeepSeek-R1 follows a "progressive separation" pattern: it starts with the highest similarity(>0.9) but separates domains continuously layer by layer, ending with distinct boundaries without the terminal re-integration seen in Qwen.

## D.3. Comparative Metrics in Critical Layers

As shown in Table 7 , Quantitative analysis of Routing Specialization $S_{spec}^{(l)}$ across critical layers exposes a fundamental divergence in resource allocation strategies.First, at the first layer, Qwen-Thinking models immediately separate the Medical domain (score 0.96), effectively reserving specific experts right from the start. In contrast, DeepSeek and GLM start with almost zero specialization (<0.10), meaning they treat all inputs equally at the beginning. Second, at the last layer, the "Thinking" training causes Qwen to become extremely specialized for complex tasks like Medical (score soaring to 1.90), much higher than its Instruct version (0.82) or other models ( 0.7). This suggests that Qwen's "Thinking" process relies greatly on dedicating the final layers to specific hard problems, whereas DeepSeek maintains a more balanced approach throughout.

*Table 7.* Routing Specialization ($S_{spec}$) across critical layers.

| MODEL | LAYER | MATH | SCIENCE | MEDICAL | MEDICAL2 | KNOWLEDGE | CODE | LEGAL | CODE2 | FINANCE |
|---|---|---|---|---|---|---|---|---|---|---|
| QWEN3-30B-INSTRUCT | LAYER 0 | 0.27 | 0.31 | 0.45 | 0.45 | 0.15 | 0.15 | 0.33 | 0.21 | 0.16 |
| | LAYER 24 | 0.68 | 0.82 | 0.85 | 0.85 | 0.54 | 0.35 | 0.96 | 0.58 | 0.46 |
| | LAYER 47 | 0.41 | 0.61 | 0.82 | 0.82 | 0.33 | 0.26 | 0.67 | 0.42 | 0.23 |
| QWEN3-30B-THINKING | LAYER 0 | 0.19 | 0.31 | 0.96 | 0.96 | 0.13 | 0.25 | 0.36 | 0.22 | 0.20 |
| | LAYER 24 | 0.76 | 0.86 | 0.85 | 0.85 | 0.62 | 0.62 | 0.98 | 0.78 | 0.53 |
| | LAYER 47 | 0.31 | 0.76 | 1.90 | 1.89 | 0.39 | 0.63 | 0.81 | 0.59 | 0.40 |
| QWEN3-235B-THINKING | LAYER 0 | 0.30 | 0.30 | 0.69 | 0.69 | 0.26 | 0.26 | 0.31 | 0.32 | 0.32 |
| | LAYER 47 | 0.55 | 0.52 | 0.57 | 0.57 | 0.42 | 0.51 | 0.67 | 0.60 | 0.60 |
| | LAYER 93 | 0.33 | 0.32 | 0.81 | 0.81 | 0.28 | 0.32 | 0.33 | 0.32 | 0.33 |
| DEEPSEEK-R1-0528 | LAYER 0 | 0.06 | 0.05 | 0.06 | 0.06 | 0.02 | 0.03 | 0.05 | 0.03 | 0.03 |
| | LAYER 29 | 0.22 | 0.29 | 0.31 | 0.32 | 0.12 | 0.19 | 0.28 | 0.26 | 0.28 |
| | LAYER 57 | 0.50 | 0.38 | 0.68 | 0.68 | 0.15 | 0.25 | 0.35 | 0.19 | 0.18 |
| GLM-4.6 | LAYER 0 | 0.09 | 0.11 | 0.15 | 0.15 | 0.05 | 0.10 | 0.11 | 0.08 | 0.09 |
| | LAYER 44 | 0.43 | 0.41 | 0.49 | 0.49 | 0.19 | 0.39 | 0.58 | 0.37 | 0.44 |
| | LAYER 88 | 0.37 | 0.31 | 0.72 | 0.72 | 0.10 | 0.35 | 0.45 | 0.20 | 0.21 |

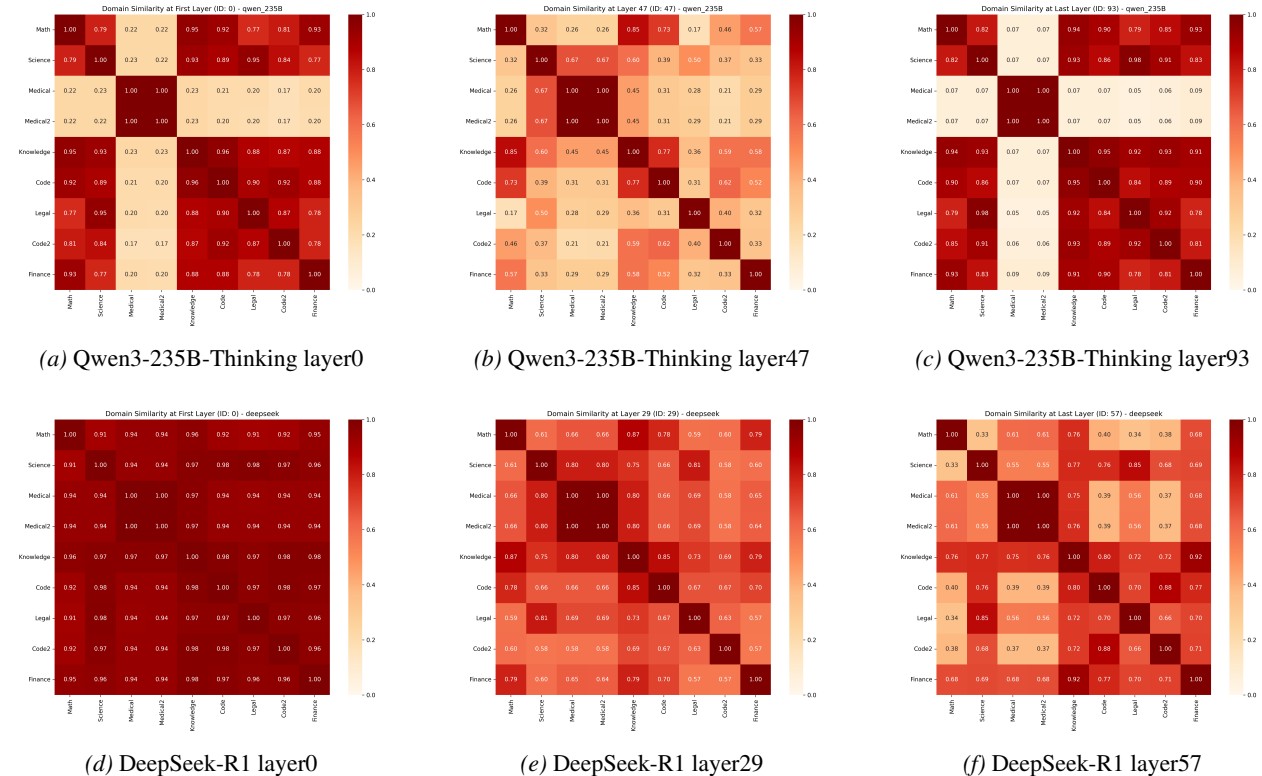

*(a)* Qwen3-235B-Thinking layer0  *(b)* Qwen3-235B-Thinking layer47  *(c)* Qwen3-235B-Thinking layer93

*(d)* DeepSeek-R1 layer0  *(e)* DeepSeek-R1 layer29  *(f)* DeepSeek-R1 layer57

*Figure 16.* **Cross-domain expert similarity heatmaps at initialization, core, and terminal layers.** The visualization compares the routing dynamics of **(a-c) Qwen3-235B-Thinking** and **(d-f) DeepSeek-R1**. While Qwen demonstrates a "Re-integration" mechanism in the final layer (except for the isolated Medical domain), DeepSeek displays a continuous "Progressive Separation" trend from a unified start to a divergent end.

## E. DBES: Domain Bench of Expert Specialty

Detailed database statistics is shown in Table 8 and the whole database is on Huggingface.

*Table 8.* Domain Bench of Expert Specialty(DBES)

| Partition | Domain | Samp. | Source | Type |
|---|---|---|---|---|
| AIME 2025 | Math | 30 | Balunovi et al. (2025) | Logical Reasoning |
| AllenAI SciQ (Val) | Science | 1000 | Johannes Welbl (2017) | Scientific Knowledge |
| BigBio MedQA (Dev) | Medical | 1623 | Jin et al. (2021) | Professional Exam |
| BigBio MedQA (Test) | Medical2 | 1815 | Jin et al. (2021) | Professional Exam |
| CAIS HLE | Knowledge | 1,600 | Phan et al. (2025) | Multi-subject Knowledge |
| LiveCodeBench (Test) | Code | 400 | Naman Jain (2024) | Code Generation |
| Nguha LegalBench | Legal | 1,600 | Guha et al. (2023) | Legal Reasoning |
| Princeton SWE-bench (Test) | Code2 | 500 | Jimenez et al. (2024) | Software Engineering |
| Yale-FinanceMath (Val) | Finance | 200 | Zhao et al. (2024b) | Financial Math |

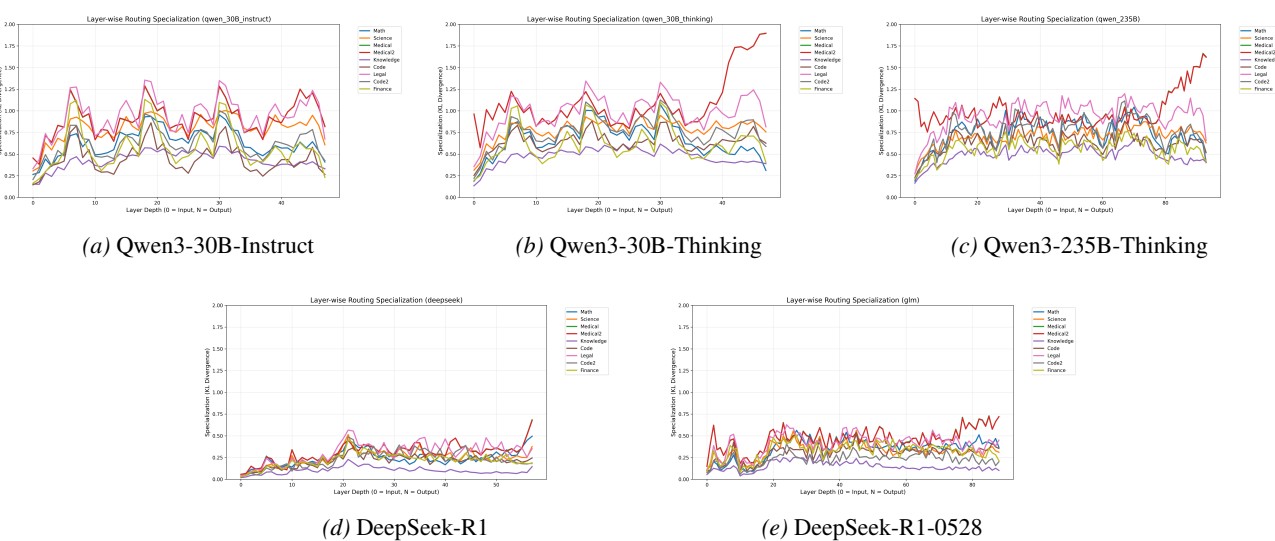

*(a)* Qwen3-30B-Instruct        *(b)* Qwen3-30B-Thinking        *(c)* Qwen3-235B-Thinking

*(d)* DeepSeek-R1        *(e)* DeepSeek-R1-0528

*Figure 17.* **Layer-wise routing specialization** ($S_{spec}$). **(a, d, e)** Standard models (Instruct/Base) follow a convergence pattern, declining in final layers for feature integration. **(b, c)** CoT-finetuned models exhibit a distinct "Terminal Surge", sustaining high specialization for reasoning. Note the magnitude gap: Qwen architectures show significantly stronger expert isolation compared to DeepSeek and GLM.

# F. Alternative Perspectives on Specialization versus Generalization

In this paper, we emphasize the specialization of MoE models, but we could not ignore the problem of exploring generalized artificial intelligence. Thus, we explore the tension and balance between specialization and generalization in machine learning models from diverse viewpoints in this part.

## F.1. The Human Understanding of Knowledge Hierarchies

In human cognition, knowledge is not stored in isolation but is organized into complex, hierarchical structuresoften conceptualized as "knowledge trees." This framework comprises:

- **A Broad Foundational Base (General Knowledge):** This includes fundamental logic, common sense, and cross-domain principles (e.g., mathematics, physical laws). These elements form the foundational framework for understanding and learning new concepts.

- **Deep, Specific Branches (Specialized Knowledge):** These are detailed, refined clusters of knowledge within specific domains (e.g., medicine, law, programming). They depend on the foundational base and are densely interconnected with other branches within the same domain.

- **Dynamic Growth and Interconnection:** New knowledge can either deepen an existing branch (specialization) or create novel connections between different branches, leading to interdisciplinary insightsa form of advanced generalization.

The development of a human expert typically involves progressing from a broad, general education to increasing depth within a chosen branch. This structure suggests that genuine "intelligence" may require a similarly scalable and deepenable hierarchical knowledge architecture.

## F.2. The Imperative for Generalization in Large Language Models

For Large Language Models (LLMs), robust generalization capability is a core value and fundamental necessity, primarily due to the following reasons:

- **Handling Open-Domain Tasks:** The distribution of real-world problems and queries is immensely wide. Models must comprehend and respond to topics, phrasings, or task combinations not explicitly seen during training.

- **Functioning as Foundational Models:** Their intended role is often as a starting point for downstream applications. Broad coverage and basic competencies in logic and language understanding (generalization) enable rapid adaptation to diverse specific needs via lightweight methods like fine-tuning or prompt engineering.

- **Fostering Emergent Abilities:** Many complex reasoning, analogical, and creative capabilities often "emerge" only after a model achieves sufficient scale and breadth of general knowledge. Premature overspecialization can potentially stifle this latent potential.

Thus, generalization ensures a model's flexibility, robustness, and viability as a foundational platform.

## F.3. The Value of Specialization for Downstream Applications

While generalization provides the foundation, specialization is crucial for translating model capabilities into practical value:

- **Enhancing Accuracy and Reliability:** In high-stakes domains like healthcare, law, or finance, model outputs must adhere to highly specialized norms, terminology, and factual standards. Specialized fine-tuning significantly reduces domain-specific "hallucinations" and commonsense errors.

- **Optimizing Efficiency and Cost:** A model optimized for a specific task (e.g., customer service, code generation) typically offers faster inference, lower computational resource requirements, and a more precise understanding of domain-specific intent.

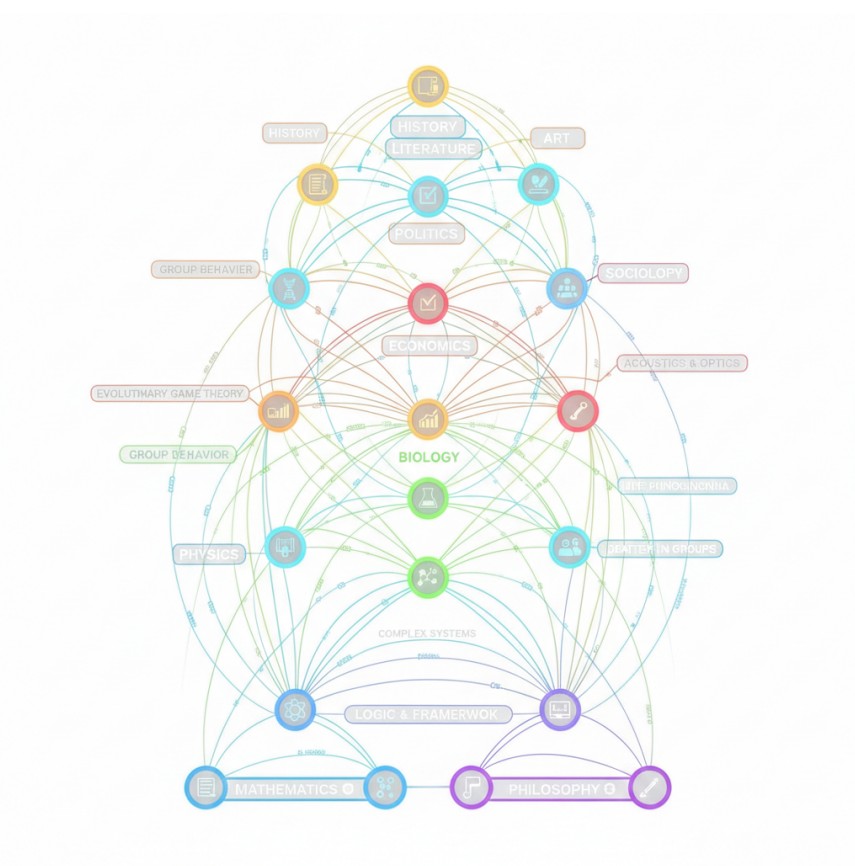

*Figure 18.* **Hierarchical Interconnectivity in Human Knowledge Graphs.** Mastery of a subject is rarely achieved in isolation; rather, it emerges from a dense, scalable, and deepenable network of interconnected domains. This suggests that "genuine intelligence" in artificial systems necessitates a similarly hierarchical architecture, moving beyond isolated expert activation toward an integrated, synergistic knowledge manifold.

- **Enabling Deep Integration:** Specialized models can more effectively understand and invoke domain-specific tools, APIs, or databases, becoming seamless automated components within professional workflows.

- **Meeting Compliance and Security Requirements:** Industry-specific standards for data handling, security, and auditability are often best addressed through controlled and verifiable specialized model versions.

Consequently, downstream applications frequently follow a "general-then-specialized" pathway: leveraging a powerful generalist model to comprehend the task and context, then applying specialization techniques to elevate the output to production-grade requirements in terms of precision, reliability, and dependability.

**Summary**

Specialization and generalization exist not as opposites but as points on a continuum. An ideal artificial intelligence system might mirror the human knowledge hierarchy, possessing a broad and solid general foundation while maintaining the ability to efficiently grow or activate deep, specialized branches as needed. Current technological approacheswhether scaling up models to enhance general capabilities or injecting expert knowledge through fine-tuning and retrieval-augmented generationare all engaged in exploring the optimal equilibrium point between these two poles.

