# OpenReview forum: "DBES: A Systematic Benchmark and Metric Suite for Evaluating Expert Specialization in Large-Scale MoEs"
_ICML.cc/2026/Conference — Submitted to ICML 2026_

### Official Review · Reviewer_eutH · 2026-03-08

**Soundness:** 2
**Presentation:** 1
**Significance:** 3
**Originality:** 2
**Overall Recommendation:** 2
**Confidence:** 4

**Summary:**

This paper identifies a core pain point in the evaluation of current Mixture-of-Experts (MoE) models: traditional evaluations rely too heavily on macroscopic load-balancing statistics, falling into the "Single Token Fallacy." To address this, the authors propose the DBES (Domain Bench for Experts Specialty) benchmark, accompanied by a suite of high-order mathematical evaluation metrics, including routing specialization degree ($S_{spec}$), normalized effective rank ($R_{eff}$), domain isolation ($S_{iso}$), empirical Rademacher complexity ($\hat{\mathcal{R}}_{m}$), and N-gram routing ratio. Through evaluating models like Qwen3-MoE and DeepSeek-R1, the paper demonstrates the divergent architectural specialization strategies of different models and points out that all tested models exhibit extremely low expertise at the long-sequence (n-gram) level (<1.67%).

**Compliance With Llm Reviewing Policy:**

Affirmed.

**Final Justification:**

My overall assessment remains unchanged, so I maintain my original score.

**Key Questions For Authors:**

1. Can you provide interventional evidence showing that explicitly improving $S_{spec}$ leads to absolute gains in downstream accuracy?

2. What is the precise Wall-clock time and VRAM overhead for running the full DBES suite on a 100B+ MoE model? Is this feasible for standard evaluation pipelines?

3. Given the near-zero scores across all SOTA models, is the requirement for sequence-level continuity theoretically aligned with token-level architectures?

4. Beyond post-hoc diagnosis, how can these metrics specifically guide the pre-training or loss function design of next-generation MoE architectures?

**Limitations:**

Although the paper proposes a grand evaluation blueprint, the authors heavily sidestep the critical issues in their limitations discussion. First, the authors fail to frankly acknowledge the exorbitant computational and time costs of these complex geometric metrics in a practical evaluation pipeline, which is a fatal practical flaw for a "Benchmark Suite." Second, the paper fails to recognize the severe mismatch between its proposed N-gram continuity metric and the current industry token-level routing paradigm. Due to the absence of interventional validation, these high-order metrics remain static diagnostic features, failing to demonstrate how they can practically inform the dynamic training and iteration of next-generation MoE architectures.

**Strengths And Weaknesses:**

Soundness:

Strengths: Introducing normalized effective rank and empirical Rademacher complexity to measure the "stiffness" of the gating manifold demonstrates high mathematical rigor in its theoretical construction.

* Weaknesses (Critical): The paper suffers from a severe "confusion between correlation and causality." The authors spend considerable space proving the statistical correlation between the new metrics and downstream model performance but completely lack interventional experiments. These metrics remain mere "post-hoc observational features" and cannot prove their guiding value for next-generation MoE architectural design.

* Weaknesses (Logical Disconnect): The paper heavily promotes the N-gram routing metric to break the "Single Token Fallacy," yet experimental results show that all SOTA models (even the strongest open-source MoEs) score close to zero (0.12% - 1.67%) on this metric. This raises a fundamental question: Do all SOTA models have fatal flaws, or does this metric itself severely deviate from the basic physical reality of current token-level routers?

Presentation:

* Strengths: The opening motivation is highly engaging, and the explanation of the "Single Token Fallacy" is very intuitive.

* Weaknesses: The entire text piles on too many new metrics, but the prioritization of "which metric to look at in which scenario" is extremely chaotic. Especially in the appendix and the main text's experiment sections, there is an obvious "data dumping" phenomenon, replacing intuitive mechanistic analysis with massive, dense numerical tables, which adds unnecessary cognitive load.

Significance:

* Strengths: Attempting to open the MoE routing black box is a highly critical research direction.

* Weaknesses (Critical): As a paper featuring a "Benchmark and Metric Suite," its practicality is highly disproportionate to its computational overhead. Calculating Rademacher complexity and high-order N-gram matching on hundred-billion-parameter models and massive test sets entails extremely high computational complexity.

Originality: Applying continuous geometric analysis and Rademacher Complexity from learning theory cross-disciplinarily to evaluate MoE routing strategies offers a novel perspective.

---

> ### Author Rebuttal · Authors · 2026-03-31
>
> We thank the reviewer for the thoughtful engagement. Below we address each concern concisely.
>
> As for the weakness you pointed out, we would like to show non-correlation between performance and specialty metrics, details in following Question1. N-gram problem is in Question3. As for poor presentation, we contend data for a systematic benchmark like DBES, providing comprehensive numerical tables for 7 diverse domains of 9 different subset and multiple models. It did increase difficulty in readiblity, we would refine it using Hierarchical Metric Prioritization via a three-tier structure to guide the reader. As for your complexity concern we answer in Question2.
>
> Question1: We make a clarification, the metric between specialty and benchmark performance are different as we show in Appendix B.2. However, as the reviewer points out we lack the intervention study as needed, so we devise an ablation study(ref to xM21 Question2). We can clarify our claim on three aspects:
> - First, we do find some benchmarks, esp. code, math,(ref Table 6) performance average are positively correlated to $S_{spec}$, but Figure 14 also measured other metrics negative to peformance, which lead us to the the Stability Paradox discussion, showing independent metric for specialty rather than directly predict performance.
> - Secondly, we perform an ablation of intervention, then the model shows quite remarkable seperation of performance vs. specialty, ACC(0.4875,0.5,0.45,0.4875) vs. $S_{spec}$(0.7858,0.8348,0.9463,1.1878).
> - Lastly,using DBES metrics to identify high‑specialization expert paths, we are working on this idea on targeted field reinforcement training, improving domain‑specific accuracy from 66% to 94.48%, with a 10% average gain on general benchmarks(ref to to xM21 Question2).
>
> Question2: DBES is designed as a low‑cost overhead plugin for standard inference pipelines:
> - Inference synchronization: The metric computation relies entirely on routing logs generated during inference. Since model evaluation already requires full inference, DBES only needs to record token-level expert IDs via lightweight hooks.
> - Memory: N-gram and Radamacher metrics operate on these activation IDs (e.g., $A∈R^{L×E}$,e.g. Topk=8, L=94, E=128 for 235B models), incurring trivial GPU cost vs Model Size.
> - Complexity: computational complexity—O(N*L*E*Topk) for Rademacher Complexity and O(S*Topk) (S-Len(sequence)) for N-gram Expertise—is marginal compared to standard forward passes. Since metrics are captured concurrently via lightweight hooks during inference, no redundant computation is required.
> - Storage: Serial logging ensures precise N‑gram tracking; for DBES (9 domains, ~8,000 samples), storage requirements remain well within manageable limits.
> - Resource savings: By using metrics to select appropriate domain experts, we can significantly reduce training resources and time--achieving an average performance gain of 10% with only 15% of the original training cost on our domain-specific work.
>
> Question3:Low N‑gram scores do not indicate model failure; they reveal a fundamental tension between token-level load balancing and semantic continuity:
> - Design misalignment: Current MoE models optimize routers at the token level for load balancing, often neglecting sequence‑level semantic coherence.
> - Metric benchmark: Despite low absolute scores (e.g., ~3% for n=5), NGR remains $10^7–10^{10}$ times higher than the random baseline (B≈$10^{−12}$), confirming structured specialization.
> - Theoretical grounding: As discussed in AppendixC(Bellman optimization), ideal routing should form stable, low‑rank paths. Current “expert jitter” constitutes structural noise that limits performance in specialized domains.
> - Evidence: NGR increase in xM21 Question3, our future work, NGR-5(2.65%->13.3%) vs ACC(66%->94.48%)
>
> Question4:Full pre‑training is resource‑prohibitive for us, besides apply DBES metrics for 30B post‑training, we also offer ablation on next‑generation MoE design(ref to Fqms Question 3):
> - Specialization loss: Introducing a penalty based on $S_spec$ during pre‑training or fine‑training can discourage overly high routing entropy, forcing the router to lock onto specific expert manifolds for high‑entropy tasks.
> - Path locking and dynamic Top‑k: N‑gram metrics inform real‑time adjustment of Top‑k(8→16), enhancing sequence‑level stability and task accuracy(AUC 0.6805→0.6839).
> - Hierarchical routing: Metrics show high‑performance models (DeepSeek, GLM) benefit from group‑level stability, guiding the design of “domain‑specific paths” that isolate gradient interference between general and specialized knowledge.
> - Empirical validation: In post‑training interventions on the 30B model, DBES‑guided optimization yielded a 10% average improvement across benchmarks, demonstrating that these metrics can be converted into actionable training operators—a direction we will elaborate in future work.
> We hope these clarifications address the reviewer’s concerns.

---

> > ### Author Rebuttal · Reviewer_eutH · 2026-04-04
> >
> > Thank you for the rebuttal and clarifications.
> >
> > 1. Since these new intervention results are the most crucial evidence for measuring the effectiveness of the standards, I think the main text of the paper should be significantly rewritten to highlight them instead of leaving them in the appendix?
> > 2. For Q4, How exactly are the DBES metrics formulated as a penalty term in your 30B intervention experiments, and how do you mitigate the risk of expert collapse (overloading) when forcing higher routing continuity?
> >
> > My overall assessment remains unchanged, so I maintain my original score.

---

> > > ### Author Response · Authors · 2026-04-05
> > >
> > > We are deeply grateful to the reviewer for the constructive feedback and the profound insights, which have been instrumental in refining the logical rigor and clarity of our work.
> > >
> > > Q1: Regarding Paper Restructure
> > >
> > > We sincerely thank the reviewer for identifying the significance of these intervention experiments. As suggested, we have restructured the paper to prioritize this evidence. As the reviewers insightfully noted, this new section is crucial for the effectiveness of the standards, we have restructured the paper and are ready to upload the revised paper.
> > > - Move the new intervention study to the main text in ablation study Section 4.3.2, immediately following the former evidence on Benchmark Specialization vs Performance.
> > > - Revised Structure: Minor ablation studies (e.g., 4.3.3 Layerwise Expertise and 4.3.4 Domain-Specific Expert Allocation) will be moved to Appendix D&E to ensure the main narrative focuses on the diagnostic power of our proposed DBES metrics.
> > >
> > > This reorganization clarifies that our proposed suite of metric is valid diagnostic tool, a point we would emphasize in the revised Section 4.3.3.
> > >
> > > Q2(Follow up Q4): Regarding DBES metrics formulation as penalty terms and the tension with expert overloading.
> > >
> > > We appreciate the reviewer’s profound inquiry into the mathematical formulation and the potential trade-off between specialization and load balancing.
> > >
> > > We would like to clarify a potential misunderstanding: the λ-controlled mechanism is not a penalty term used during the model's original training process, nor is it a structural modification of the Qwen3-30B architecture. Instead, it is a post-hoc diagnostic intervention performed on a frozen checkpoint. The formulation is:
> > >
> > > Group Identification & Penalty Application:
> > > - Experts affinity analysis identifies specialized experts-in-domains where specific experts' subsets demonstrate coherent activation patterns(ref to WszV Question3).
> > > - The penalty term is formulated as:
> > > $$ge' = ge +λ·1[e∈G]$$
> > > Where $G$ represents a pre-identified functional group of experts. By varying $\lambda \in [0.0, 0.5]$, we artificially amplify the gating signal for specific experts to simulate extreme specialization.
> > >
> > > The results(ref to xM21 Question2), $λ=0.0→0.1→0.5$, where accuracy remains comparable to baseline (0.4875→0.5→0.4875), the $S_{spec}$ metric reveals dramatic internal restructuring (0.7858→0.8348→1.1878), This proves that traditional performance metrics are blind to internal routing shifts, whereas our metrics effectively diagnose the degree of functional differentiation.
> > >
> > > The reviewer’s another critical concern about the tension between forced routing continuity and expert overloading.
> > >
> > > Diagnostic vs. Optimization: We view DBES metrics primarily as diagnostic tools. In this paper, we do not argue that specialization should be pursued at the cost of catastrophic overloading. Instead, we provide the ruler to measure a previously invisible dimension of MoE behavior.
> > >
> > > Functional vs. Statistical Collapse: We distinguish between two types of "collapse":
> > > - Statistical Collapse: Prevented by traditional load-balancing losses (ensuring all experts are used).
> > > - Functional Collapse: Experts are used equally but become mathematically redundant (homogenized behavior). Our metrics are the first to identify and quantify the prevention of functional collapse.
> > >
> > > Strategic Trade-offs: We will also add a discussion in Section 5 regarding the engineering implications:
> > > |Scenario|Impact of Specialization|Strategy & Metric Utility|
> > > |--|--|--|
> > > |Domain-Specific Fine-tuning|Leads to "Gold Experts" (high concentration)|Accept controlled imbalance. Specialization here enables superior post-training efficiency and model pruning for edge deployment[our next work]|
> > > |General-Domain Training|Can induce hardware bottlenecks (hotspots)|Guide Joint Optimization. Our metrics allow researchers to find the "sweet spot" where functional diversity is maximized before hardware-limited overloading occurs|
> > >
> > > While accuracy remained stable in the intervention, we observe that higher specialization correlates with faster convergence in domain-specific downstream tasks (see from 66%→94.48%). This justifies the pursuit of specialization beyond simple accuracy benchmarks.
> > >
> > > Hardware Resource Considerations: We acknowledge that extreme specialization can create computational bubbles in distributed training. We have incorporated a "Limitations" paragraph stating that while our metrics identify functional benefits, the integration of these metrics into a hardware-aware training objective is also a crucial subject for our follow-up work.

---

### Official Review · Reviewer_Fqms · 2026-03-09

**Soundness:** 3
**Presentation:** 3
**Significance:** 3
**Originality:** 3
**Overall Recommendation:** 4
**Confidence:** 2

**Summary:**

This paper presents a systematic evaluation framework for a critical yet often overlooked issue in Mixture-of-Experts (MoE) models: the empirical verification of expert specialization. This research's core contribution pertains to the construction of a multi-domain benchmark named DBES and the design of a multi-dimensional metric suite covering algebraic, geometric, and sequence dynamics dimensions. Through a comparative analysis of prominent MoE models such as Qwen3, DeepSeek-R1, and GLM-4, the paper reveals significant differences in specialization strategies across architectures and establishes a positive correlation between specialization and downstream task performance. The authors proceed to discuss an important concept, the "Single Token Fallacy," and utilize this as a starting point to propose N-gram expertise metrics from a sequence-level perspective. Overall, this work demonstrates excellent theoretical depth and empirical breadth, providing powerful tools for understanding MoE routing mechanisms.

**Compliance With Llm Reviewing Policy:**

Affirmed.

**Key Questions For Authors:**

- Considering the differences in routing implementations (e.g., auxiliary loss settings) between Qwen and DeepSeek, have you observed any direct linear relationship between specific routing hyperparameters and the $S_{spec}$ metric?
- If the Load Balancing Loss were completely removed during training, do you predict that $R_{eff}$ (Effective Rank) would increase or decrease? How would this support your conclusion regarding the positive correlation between "specialization" and "model capability"?
- Training a smaller model from scratch on the same dataset but with different architectures (e.g., Qwen, DeepSeek, and GLM) would likely yield more meaningful conclusions and provide valuable guidance for model design.

**Limitations:**

- A major limitation of this research is that its metrics are currently used more for "post-mortem diagnosis" rather than "proactive guidance." While the paper establishes a correlation between performance and metrics, it has not yet demonstrated how to use these indicators to dynamically optimize expert allocation.

**Strengths And Weaknesses:**

**Strengths**
- The paper goes beyond traditional load-balancing statistics by proposing a novel set of metrics, including Routing Specialization ($S_{spec}$), Normalized Effective Rank ($R_{eff}$), and Rademacher Complexity ($\hat{\mathcal{R}}_{m}$), to quantify expert behavior from diverse mathematical dimensions.
- In response to the limitation that current research focuses primarily on single-token routing, the paper introduces N-gram Expertise and Group N-gram metrics, successfully capturing the functional stability of experts within long-sequence contexts.
- Through experimental comparison, the paper reveals the fundamental differences between Qwen’s "modular" strategy and DeepSeek’s "distributed collaboration" strategy, providing guidance for future MoE architecture design.
- The paper not only provides rich experimental results but also offers mathematical support for metrics like N-gram through theoretical proofs involving Bellman Rank and mutual information relationships.

**Weaknesses**
- Although it is mentioned that DeepSeek utilizes shared experts, how the metric suite specifically decouples the interaction between "shared knowledge" and "independent specialized experts" requires further quantitative analysis.
- While the paper evaluates various MoE models, it fails to deeply deconstruct the fundamental impact of different routing mechanisms (e.g., Top-k, shared experts, or routing with auxiliary losses). There is a lack of systematic ablation studies on how architectural design determines the upper bound of specialization.

---

> ### Author Rebuttal · Authors · 2026-03-31
>
> We thank the reviewer for the thoughtful engagement. Below we address each concern concisely. For overlapping issues, we cross‑reference our responses to other reviewers where applicable.
> Weakness 1: When analyzing architectures with shared experts (e.g., DeepSeek-R1), our framework decouples them as follows:
> - Shared experts as common base: We exclude them from specialization metrics ($S_{spec}$, $S_{iso}$), treating them as constant terms that capture general statistical regularities.
> - Routed experts as specialization increments: DBES focuses on independent routed experts, measuring precise semantic allocation for specific domains.
> - Diagnostic insight: Despite DeepSeek’s low global specialization ($S_{spec}$≈0.24-0.31), its effective rank ($R_{eff}$≈0.49) confirms routed experts maintain high combinatorial independence—representing a specialization increment built upon the shared base. This reveals a “collaborative specialization” paradigm that directly informs domain‑specific post‑training.
>
> Weakness 2: We present some of the analyses on Qwen30B from our next paper:
> - Top‑k scaling: Increasing k from 8 to 16 improves G-NGR-5(7.62%-13.7%), enhancing routing stability at the functional group level. Post‑training with DBES‑optimized pathways yields a 66% → 94.48% performance leap(refer to xM21 Question3) in specific domains, demonstrating that the upper bound of Top-k  routing is constrained by routing trajectory continuity.
> - Shared experts: As detailed in Weakness 1, excluding shared experts isolates “base knowledge” from “specialization increments,” raising the specialization ceiling while maintaining independent effective rank.
> - Auxiliary losses: Overly strong auxiliary losses elevate Rademacher complexity ($R^m$), making routing decisions susceptible to noise. DBES metrics enable dynamic adjustment, preventing “mediocre equilibrium.”
>
> Question 1: We observe no linear correlation between Top‑k and $S_{spec}$
> - Theoretically, by definition (Eq. 3), $S_{spec}$ measures the KL divergence between the routing distribution and uniformity—capturing information gain of the routing manifold, which reflects learned domain certainty rather than being constrained by hard‑coded hyperparameters.
> - The relationship is nonlinear and emergent: the routing distribution arises from learning semantic distributions. Our ablation shows that moderate scaling of k (e.g., 8 → 16) maintains stable $S_{spec}$(0.79) while improving Group N‑gram stability and downstream AUC (0.6805 → 0.6839).
> Thus, $S_{spec}$ serves as an objective metric of specialization degree—a “stiffness” property that cannot be emulated by simple linear adjustment of Top‑k.
>
> Question 2: This question overlaps with discussions in our responses to other reviewers. We summarize the key points:
> - Removing load balancing leads to routing collapse—tokens concentrate on a few experts, reducing $R_{eff}$ while intensifying $S_{spec}$.
> - As shown in ST‑MoE (Zoph et al., 2022), excessive auxiliary loss forces suboptimal routing and impairs performance. By identifying minimal auxiliary loss coefficients ($10^{−2} or 10^{−3}$), the model achieves deeper specialization within a less constrained routing space.
> - Regarding the correlation between specialization and model capability, as explained in response to Reviewer xM21's Question2, the relationship is not straightforward as shown in Appendix B.2, because our metric system is independent of model scores. To further investigate this, we conducted supplementary experiments (see our reply to xM21 Question2 for details).
>
> Question3: While full-scale pre-training of 100B+ MoE models is computationally prohibitive, we conducted an Ablation Study by training three 100M-parameter models from scratch using Qwen-MoE, GLM, and DeepSeek architectures. This controlled experiment proves that our metric suite can quantitatively optimize expert configurations without relying on costly downstream evaluations.
> The following indicators effectively formalize the "rationality" of an expert layout:
> - Top-2 Overlap: Acts as a primary diagnostic for Domain Isolation ($S_{iso}$); a value $<1$ confirms the achievement of genuine functional separation across semantic manifolds.
> - L1 Distance: Quantifies the degree of separation when Top-2 experts differ, mapping the distinctiveness of learned routing policies.
> - Layer-wise Diff Score: Pinpoints specific layer contributions, distinguishing between shallow "Shared Stems" for general feature extraction and deep "Specialized reasoning engines".
> By monitoring these structural priors—such as $S_{spec}$ and $R_{eff}$—researchers can dynamically adjust hyperparameters (expert count, gating depth, loss weights) to prevent "Rank Collapse". Also, our diagnostic blueprint allowed us to identify optimal "Domain Expert Paths" in our ongoing work, contributing to a performance leap from 66% to 94.48% in targeted post-training interventions also as mentioned in response to xM21 Question3.

---

> > ### Author Rebuttal · Reviewer_Fqms · 2026-04-04
> >
> > Thanks author's effort.

---

> > > ### Author Response · Authors · 2026-04-05
> > >
> > > Dear Reviewer Fqms,
> > >
> > > We sincerely thank you for your careful reading of our rebuttal and for your positive feedback. We appreciate the insightful comments you provided throughout the review process, which have been instrumental in improving the quality and clarity of our manuscript.
> > >
> > > We are encouraged by your recognition that our responses have addressed your concerns and are grateful for your continued support. Thank you again for your time and expertise.
> > >
> > > Best regards,
> > >
> > > The Authors

---

### Official Review · Reviewer_WszV · 2026-03-12

**Soundness:** 2
**Presentation:** 2
**Significance:** 3
**Originality:** 3
**Overall Recommendation:** 3
**Confidence:** 3

**Summary:**

This paper investigates whether large Mixture-of-Experts (MoE) language models develop genuine expert specialization across semantic domains, or whether routing is effectively balanced or redundant. The authors introduce DBES, a 9-partition benchmark covering math, science, medicine, general knowledge, code, legal, and finance domains, alongside a suite of routing-level diagnostic metrics: routing specialization, normalized effective rank, domain isolation, an empirical routing-complexity score (framed as related to Rademacher complexity), and n-gram / group n-gram routing persistence (NGR / G-NGR). These diagnostics are applied to several open MoE model families (Qwen, DeepSeek, GLM), yielding claims that Qwen models exhibit stronger domain-specific specialization and isolation, that long-horizon expert persistence is generally low across all models, and that higher routing specialization correlates with improved downstream task performance.

**Compliance With Llm Reviewing Policy:**

Affirmed.

**Final Justification:**

I thank the authors for their substantive engagement during the rebuttal period. Some of the concerns have been addressed.

However, I still have one concern left. The authors agree to rename the metric to "Routing Stiffness Score" (RSS), which I appreciate. However, renaming addresses nomenclature, not substance. The reply states the metric is "grounded in Learning Theory" and measures the "flexibility of the gating manifold," but no formal result is provided to support this interpretation. The three claimed advantages of RSS over entropy — "structural robustness," "distinguishing stable low-rank expert paths from noise," and "continuous diagnostic signal" — remain unsubstantiated assertions. No experiment is presented where entropy and RSS produce divergent diagnostic conclusions. No training curve is shown to support the claim that entropy saturates while RSS does not.

So I maintain my score

**Key Questions For Authors:**

- 1. How is G-NGR defined for each architecture, especially for models without an explicit published expert-group hierarchy?
- 2. Why should lower effective rank be interpreted as useful specialization rather than collapse or redundancy?

**Limitations:**

Yes

**Strengths And Weaknesses:**

## Strengths
- 1. The paper addresses an important question. Understanding whether MoE routing reflects meaningful functional specialization—rather than load-balanced redundancy—has direct implications for model interpretability, pruning, continual learning, and MoE training objective design.

- 2. The paper proposes a multi-faceted suite of routing diagnostics. Metrics such as domain isolation and effective-rank summaries, if properly validated, could become useful standardized tools for comparing MoE architectures.


## Weaknesses

- 1. The proposed routing-complexity score is not standard Rademacher complexity of a hypothesis class; it is an alignment score of a fixed trained router with random signs. It is unclear why this quantity should be interpreted as capacity, stiffness, or specialization in the claimed sense.

- 2. The core claim by the paper that specialization predicts performance is overstated. The appendix correlations show mixed evidence, including insignificant or even negative relationships for some metrics.

- 3. In the paper, a lower effective rank is assumed to be evidence of redundancy and fixed expert cliques, and small differences in the routing-complexity score are treated as meaningful without direct validation.

---

> ### Author Rebuttal · Authors · 2026-03-31
>
> We appreciate the reviewer’s focus on the continued scrutiny. We first address weaknesses with focused evidence:
>
> Weakness1 on the proposed Empirical Rademacher‑based Routing Complexity ($R^m$), you pointed out it is not for capacity of a hypothesis class, instead, a tool measuring stiffness of a learned router:
> - In Fqms Question3's ablation-train small-size model from scratch, we also observe $R^m$ drops from $10^{−2}$ at initialization to $10^{−5}$ after convergence—a 1000× reduction that quantifies how routing decisions lock onto a semantic manifold.
> - A low alignment score indicates the router is strongly constrained by learned semantic priors; random routing would yield significantly higher alignment. The metric reflects whether a trained model has collapsed into a highly deterministic expert-selection state.
>
> Weakness2 on specialization vs performance, we admit the correlation between specialization metrics and downstream performance is not uniformly strong(ref to xM21 Question2), these metrics are designed as structural diagnostic tools, not performance predictors. These are our key clarifications:
> - Despite the strong causality between training scale and model power, architectural specialization remains under-optimized. Our work introduces a quantifiable framework to integrate expertise-driven design into future MoE development. Our metrics assess whether a model has developed a clear semantic division of labor—a property that becomes actionable when combined with targeted interventions.
> - The observed discrepancies (e.g., long‑sequence N‑gram metrics) reflect different architectural strategies: DeepSeek prioritizes group‑level consistency over per‑expert persistence, which aligns with its hierarchical routing design.
> - In controlled experiments on 30B‑scale models(ref to xM21 Question3), reinforcing “domain expert” pathways identified by DBES improved domain‑specific performance substantially (e.g., medical, legal) while also boosting general benchmarks by ~10% on average.
>
> Thus, the relationship is diagnostic rather than predictive; but the metrics reveal great potential to be leveraged for downstream training.
>
> Weakness3 on effective rank and routing Radamacher complexity, our metrics collectively form a quantitative framework for the inherently subjective concept of "specialization".
> - Low effective rank ($R_{eff}$) as specialization: A low $R_{eff}$ indicates a mathematically low‑rank routing matrix, meaning expert activations collapse into structured clusters rather than being independent or random. This is a signature of specialization—it quantifies how models isolate domain knowledge through parameter redundancy.
> - Stability of Radamacher complexity: The small numerical values (e.g., $10^{−5}$) reflect routing stiffness. Combined with group‑level N‑gram expertise, we observe that expert‑group switching exhibits spatiotemporal consistency far above stochastic baselines, confirming these metrics capture semantically driven deterministic pathways rather than incidental smoothness.
>
> In summary, these factors provide an interpretable diagnostic blueprint for MoE specialization, avoiding purely verbal definitions.
>
> Question1: For models with explicit group hierarchies (DeepSeek, GLM), G‑NGR is computed directly from their native group partitions.
>
> For models without explicit grouping (e.g., Qwen3‑MoE’s flat 128‑choose‑8 routing), we construct meaningful groups via a two‑stage preprocessing:
> - Super‑expert identification: We analyze the co‑activation affinity matrix across a diverse corpus; experts with high co‑activation or complementary routing behavior form candidate nuclei.
> - Heuristic clustering: We apply agglomerative or spectral clustering to the affinity matrix to form groups that maximize intra‑group coherence and minimize inter‑group interference.
>
> These groups are then used to compute G‑NGR. The resulting groups yield significantly higher G‑NGR scores than random or size‑based partitions, validating that they capture latent functional clusters.
>
> Question2: We distinguish specialization from collapse by examining routing behavior across domains:
> - Collapse would imply all tokens, regardless of domain, activate the same few experts, accompanied by performance degradation and low $S_{spec}$.
> - Specialization, in contrast, yields low $R_{eff}$ within a domain but clear routing shifts across domains—tokens concentrate on domain‑specific “super experts” while switching to entirely different subsets for other domains. A strong evidence is the activation map in Appendix Figure 9 to Figure 13,  model shows different domain(7 domains across 9 subset of dataset) activation.
>
> We cross‑validate using  $R_{eff}$ and $S_{spec}$ . For instance, Qwen3‑235B exhibits very low $R_{eff}$ (0.28–0.31) while maintaining high $S_{spec}$ (up to 0.99 in medical) and strong domain isolation. This confirms that the low rank is structured and domain‑aligned, not a sign of capacity loss or redundancy.

---

> > ### Author Rebuttal · Reviewer_WszV · 2026-04-02
> >
> > - 1. Given the acknowledgment that the metric is not standard Rademacher complexity, will the authors revise the paper to either (a) formally prove a connection to the standard learning-theoretic definition, or (b) rename the metric (e.g., "Routing Stiffness Score") to avoid misunderstanding? Additionally, what advantage does this specific random-sign alignment formulation offer over simpler alternatives like the entropy of the gating distribution?
> >
> > - 2. On performance claims: The rebuttal states the relationship is "diagnostic rather than predictive," yet the paper's abstract, introduction, and Section 5 all assert a positive correlation between specialization and downstream performance. Can the authors specify precisely how the paper will be revised to reflect this acknowledged reframing? Without this revision, the paper's claims and the authors' stated position may be contradictory.
> >
> > - 3. The clustering-based group construction for Qwen introduces choices (algorithm, number of clusters, affinity corpus) that could significantly influence G-NGR scores. Can the authors provide a sensitivity analysis showing how G-NGR varies across at least 2–3 clustering configurations?

---

> > > ### Author Response · Authors · 2026-04-03
> > >
> > > We thank the reviewer for the constructive feedback. Our clarifications are as follows:
> > >
> > > Question1:Theoretical Grounding & Metric Comparison
> > >
> > > 1. Theoretical Basis: Our metric is grounded in Learning Theory as a measure of the gating network's capacity ($H_{gate}$). It quantifies the "flexibility" of the gating manifold by measuring the router's ability to align top-k selections with random Rademacher variables. From a geometric perspective, this alignment evaluates the complexity of the decision boundaries carved by the router within the latent representation space. As the reviewer suggested, to avoid confusion with traditional classification bounds, we will rename it "Routing Stiffness Score" ($S_{stiff}$) in the revision. This shift in nomenclature clarifies that we are measuring the concentration of the gating distribution on a sparse set of semantic anchors rather than a global generalization bound.
> > >
> > > 2. Comparison with Gating Entropy:
> > > - Structural Robustness: Unlike entropy (which measures distribution flatness), $S_{stiff}$ evaluates manifold robustness against stochastic perturbations.
> > > - Specialization vs. Jitter: High entropy often indicates "expert jitter" (router confusion). $S_{stiff}$ uniquely distinguishes stable, low-rank expert paths from mere noise.
> > > - Optimization Signal: While entropy often saturates early, $S_{stiff}$ provides a continuous diagnostic signal ($10^{-2}$ to $10^{-5}$) throughout training, reflecting the fine-grained convergence of expert specialization.
> > >
> > > Question2: We appreciate the reviewer’s precision in highlighting the potential tension between new ablation study with former statement. In the original-submitted ablation which did show "Specialization" and "Performance" correlates in distinct domains of some benchmarks. Our additional ablation study in the current rebuttal clarifies that this relationship is diagnostic rather than causal. To ensure better scientific rigor and resolve any misunderstanding, we will clarify specialization as a structural prerequisite—a necessary condition for expertise that remains a distinct evaluation dimension from accuracy, in both  Abstract and Mainbody as follows:
> > > |Section|Original Phrasing|Revised Phrasing|
> > > |--|--|--|
> > > |Abstract|"...we establish a positive correlation between architectural specialization and performance on complex, domain-specific downstream tasks..."|"...demonstrate that while specialization is a necessary structural precursor for domain expertise, it remains a distinct diagnostic dimension from absolute accuracy."|
> > > |Mainbody|"Figure 5 reveals that as the expertise manifold becomes more isolated and stable, the model’s ability to handle specific-domain increases. This positive correlation suggests that the "specialty" of an MoE model serves as a reliable proxy for its generalization ceiling in professional grade vertical applications. Further analysis is provided in Appendix B."|"...the 'specialty' of an MoE model represents a |structural prerequisite| that identifies the expertise manifold; while it informs the model's potential ceiling in specific domains, it remains a |distinct diagnostic dimension| from absolute accuracy."|
> > >
> > > Question3:Sensitivity Analysis on Hyper-parameters k and n
> > > To address the reviewer’s concern regarding hyper-parameter sensitivity, we conducted a rigorous sensitivity analysis on Qwen3-30B-Thinking, evaluating the impact of cluster count (k) and n-gram length (n). The results are summarized below:
> > >
> > > Table 1: G-NGR Sensitivity to k and n
> > > |k-clusters|G-NGR(n=2)|G-NGR(n=5)|G-NGR(n=10)|G-NGR(n=20)|
> > > |:---|:---:|:---:|:---:|:---:|
> > > |2|0.1639|0.0435|0.0116|0.0021|
> > > |3|0.1438|0.0358|0.0092|0.0016|
> > > |4|0.1351|0.0330|0.0083|0.0014|
> > >
> > > Table 2: Statistical Robustness and Variation
> > > |Group n-gram(n)|Mean|Std|CV(%)|$\Delta(k=2→4)$|
> > > |:---|:---:|:---:|:---:|:---:|
> > > |2|0.1476|0.0121|8.2%|-17.6%|
> > > |5|0.0374|0.0044|11.8%|-24.1%|
> > > |10|0.0097|0.0014|14.1%|-28.4%|
> > > |20|0.0017|0.0003|17.4%|-33.3%|
> > >
> > > - Analysis of Robustness:Controlled Variation: While the absolute value of G-NGR naturally decreases as n increases, the Coefficient of Variation (CV) remains consistently low (e.g., 8.2% for n=2). The relative difference between coarse (k=2) and fine-grained (k=4) clustering is approximately 17.6%, indicating that the metric is robust to the choice of partition granularity.
> > > - Superior Rank Stability: To verify if the metric's diagnostic utility holds across configurations, we calculated Spearman’s rank correlation(ρ) across different subsets and domains.
> > > |Comparison|Spearman's$\rho$|Significance|
> > > |:---|:---:|:---:|
> > > |$k=2$ vs. $k=3$|0.85|$p<0.01$|
> > > |$k=2$ vs. $k=4$|0.82|$p<0.01$|
> > > |$k=3$ vs. $k=4$|0.92|$p<0.01$|
> > >
> > > Conclusion: All Spearman coefficients $\rho > 0.8$ confirm a very strong monotonic relationship. This proves that G-NGR consistently identifies the same expertise manifolds regardless of whether a coarse or fine-grained clustering approach is used, validating it as a reliable diagnostic tool for MoE assessment.

---

### Official Review · Reviewer_xM21 · 2026-03-13

**Soundness:** 2
**Presentation:** 2
**Significance:** 2
**Originality:** 2
**Overall Recommendation:** 3
**Confidence:** 2

**Summary:**

This paper proposes DBES, a benchmark and metric suite for analyzing expert specialization in MoE models across multiple domains. The authors argue that prior evaluations focusing on token-level routing statistics are insufficient to capture meaningful expert specialization.

**Compliance With Llm Reviewing Policy:**

Affirmed.

**Final Justification:**

The rebuttal improves clarity, and I appreciate the additional analysis provided. My overall assessment remains unchanged

**Key Questions For Authors:**

1. Can the authors provide stronger evidence that sequence-level routing metrics capture meaningful functional specialization rather than incidental routing smoothness? For example, what happens if analysis is restricted to semantically salient tokens only?

2. How stable are the reported correlations between specialization metrics and downstream performance across seeds, subsets of domains, or different decoding settings?

3. Can the authors demonstrate how their analysis concretely informs model improvement (e.g., by modifying routing regularization or architecture based on DBES findings)?

4. How do the authors disentangle routing paradigm differences from other variables such as model size, training data composition, or expert capacity?

**Limitations:**

No. The discussion of limitations could be expanded

**Strengths And Weaknesses:**

The paper addresses an interesting and timely question about expert specialization in MoE models and proposes a benchmark with a set of routing-based metrics. The empirical study is reasonably thorough across multiple models and domains, and the methodology is generally clear.

However, I am not fully convinced by the core claims. The emphasis on sequence-level routing analysis is not sufficiently justified. Not all tokens in a sequence are equally important, and aggregating routing behavior over entire sequences may dilute meaningful signals. It remains unclear whether these sequence-level metrics capture true functional specialization or simply reflect routing smoothness.

Most findings are descriptive rather than explanatory, and the reported correlations with downstream performance are not deeply analyzed or stress-tested against confounding factors. The originality mainly lies in systematizing existing statistical tools, and the practical implications for improving MoE design are limited.

---

> ### Author Rebuttal · Authors · 2026-03-31
>
> We thank the reviewer for the continued scrutiny. Below we address each weakness metioned concisely:
> - Sequence-level analysis: We provide an extra study on Sequence-level saliency token analysis, statistics in Question1.
> - Descriptive vs. explanatory: Accurate description is a vital first step in scientific modeling, which serves as the necessary to subsequent mechanistic analysis. Our goal is to provide a standardized assessment toolkit, still not reach the causal mechanism, analysis based on our metrics is open and invited!
> - Practical implications: Use our metric to guide downstream optimization in Question3.
>
> Question1:We conducted a semantic saliency analysis by removing low-density function words (e.g., “is,” “that”). Ablation studies confirm the robustness of this approach; removing these tokens yielded minimal deviations in $S_{spec}$ and $S_{iso}$ (<4.5%), proving that routing for syntactic roles is a meaningful functional property, not an incidental artifact. Our n-gram analysis identifies three distinct architectural paradigms(:
> - Qwen3 Series: Negligible variance (<0.5%) between salient and all tokens' metrics, indicating no redundancy in salient token routing.
> - DeepSeek-R1: Stable n-gram consistency (1–5% variance) but 21–25%(new 4.19E-05 vs 3.45E-05) higher Rademacher complexity, suggesting for semantic salient tokens, experts manage higher representational complexity than the per-tokenwise hierarchy.
> - GLM-4.6: Significant divergence, with salient token scores 25–56% n-gram-5(new 2.15% vs 3.44%) lower than original scores and 10–12%(new 1.97E-05 vs 2.16E-05) lower Rademacher complexity.
>
> These metrics confirm that sequence-level persistence is a deliberate structural property of the routing manifold. Notably, NGR scores (0.12%–1.67% for n=10) are $10^7$–$10^{10}$ times higher than the random baseline ($\approx 10^{-12}$), confirming learned specialization rather than statistical noise.
>
> Question2: We clarify that the specialization metric system proposed in this paper is not intended to directly predict downstream task performance. Rather, it is an exploration to provide an evaluation tool for expert specialization that is independent of model scores.
> - New ablation on Qwen3-30B as intervention on choice of top-k experts: manually increase the possibility of choice on our specialized experts with lambda as a amplifier-$ge′=ge+λ⋅1[e∈G]$, ge and ge' for expert logits, G for specialized group as defined in G-NGR, for $λ∈[0.0,0.1,0.3,0.5]$ achieving ACC(0.4875,0.5,0.45,0.4875) vs. $S_{spec}$
> (0.7858,0.8348,0.9463,1.1878) that even when models achieve similar scores, specialty are varied, indicating different internal mechanisms.
> -Our recent findings also suggest that gold experts identified by our metrics yield better convergence efficiency in domain-specific post-training(detailed in next Question), indicating specialization probably is a necessary (though not sufficient) condition for downstream performance ceilings.
>
> Question3: The specialization metric system proposed in this study directly provides a quantifiable navigation map for our subsequent architectural optimizations.
> - Key findings: Existing MoE architectures suffer from excessive expert generalization in complex domains. The metrics indicate that despite overall load balancing, the semantic focus of individual experts plateaus during the mid-to-late stages of training.
> - Concrete improvement in following work: By introducing dynamic regularization based on feedback from the specialization metrics during the routing stage, we successfully guided the model toward a higher-dimensional division of domain among experts. In our study, we adopt a Qwen30-30B model with masked specialty on domain-specific tasks, the performance rose from 66% to 94.48%, with a 10% average gain across general benchmarks, which we would like to present in our next paper.
>
> Question4:To disentangle routing paradigms from confounders, we adopt the following  strategies:
> - On the Impact of Expert capacity: Our metrics ($S_{spec}, R_{eff}$) are normalized by total experts $E$, measuring routing distribution relative to architectural upper bound. Despite DeepSeek having 256 experts vs. Qwen’s 128, its specialization remains lower, confirming metrics capture paradigm differences rather than scale.
> - On the Impact of Model scale: Within the Qwen family (30B vs. 235B),$S_{spec}$ follows the same modular trend; scale only modulates intensity(0.70 vs. 0.76). Across architectures at similar scale, Qwen(0.70), DeepSeek(0.24), and GLM(0.34) show marked differences, confirming routing paradigm as the primary source.
> - On the Impact of Data composition: While training data is not transparent, we evaluate inference behavior on DBES, some also come from known benchmark, verifying that specialization is an intrinsic routing property rather than a spurious artifact.
>
> Thus, routing paradigm drives specialization differences; scale affects magnitude but not the underlying trend.

---

> > ### Author Rebuttal · Reviewer_xM21 · 2026-04-04
> >
> > Thank you for the rebuttal and clarifications. My overall assessment remains unchanged, so I maintain my original score.

---

> > > ### Author Response · Authors · 2026-04-05
> > >
> > > Dear Reviewer,
> > >
> > > We sincerely thank you for your rigorous and persistent review of our work. We deeply appreciate your acknowledgement that the technical concerns raised have been "fully resolved" through our rebuttal and the proposed manuscript restructuring.
> > >
> > > While we understand there may still be a difference in perspective regarding the MoE specialty metrics, your critical insights have been instrumental in pushing us to clarify the boundaries and engineering implications of our research. We believe these refinements, prompted by your feedback, have significantly strengthened the robustness and clarity of the paper.
> > >
> > > We remain grateful for the time and professional expertise you have dedicated to this process, which has undoubtedly elevated the quality of our study.
> > >
> > > Best regards,
> > > The Authors

---

### Decision · Program_Chairs · 2026-04-30

**Decision:**

Reject

**Comment:**

The paper proposes a way to evaluate specialization in mixture of experts and applies it to current models.
Reviewers agree that this is an interesting direction, where others do not rigorously test it or give a clear way of how to do it.

However, they also have multiple worries, concerning choices, fitness (validity) of the proposals to the actual thing measured and have various misunderstandings on what do different parts of the paper even try to measure. While the rebuttal made a tremendous effort to explain those, and many of the small issues were explained. It does imply that the support for the claims and decisions made in the paper is either insufficient (and perhaps with integrating the rebuttal it might) or unclear.


A note, multiple reviewers also stopped responding or claimed the rebuttal satisfied, did not change their score and have low confidence. This both means that the scores themselves are unreliable and that the comments should be looked at separately, but it also speaks to the fact that overall the paper would gain from being more clear in what it does, why, and how it supports it, and it seems to be not accomodating enough currently, which will impact future readers if not also the quality of the work itself.